# Fractals: An Eclectic Survey, Part II

**Akhlaq Husain [1] , Manikyala Navaneeth Nanda [2] , Movva Sitaram Chowdary [2,† and Mohammad Sajid [3,\***

1. Department of Applied Sciences, BML Munjal University, Gurgaon 122413, India; akhlaq.husain@bmu.edu.in
2. School of Engineering & Technology, BML Munjal University, Gurgaon 122413, India; navaneeth.manikyala.17cse@bml.edu.in (M.N.N.); sitaram.movva.17mec@bml.edu.in (M.S.C.)
3. Department of Mechanical Engineering, College of Engineering, Qassim University, Buraydah 51452, Saudi Arabia
* Correspondence: msajd@qu.edu.sa
† Current address: IMSE, College of Engineering & Computer Science, University of Michigan, Dearborn, MI 48126, USA.

**Abstract:** Fractals are geometric shapes and patterns that can describe the roughness (or irregularity) present in almost every object in nature. Many fractals may repeat their geometry at smaller or larger scales. This paper is the second (and last) part of a series of two papers dedicated to an eclectic survey of fractals describing the infinite complexity and amazing beauty of fractals from historical, theoretical, mathematical, aesthetical and technological aspects, including their diverse applications in various fields. In this article, our focus is on engineering, industrial, commercial and futuristic applications of fractals, whereas in the first part, we discussed the basics of fractals, mathematical description, fractal dimension and artistic applications. Among many different applications of fractals, fractal landscape generation (fractal landscapes that can simulate and describe natural terrains and landscapes more precisely by mathematical models of fractal geometry), fractal antennas (fractal-shaped antennas that are designed and used in devices which operate on multiple and wider frequency bands) and fractal image compression (a fractal-based lossy compression method for digital and natural images which uses inherent self-similarity present in an image) are the most creative, engineering-driven, industry-oriented, commercial and emerging applications. We consider each of these applications in detail along with some innovative and future ready applications.

**Keywords:** fractals; iterated function system; fractal landscapes; fractal antenna; fractal image compression; fractal batteries; fractal capacitors; fractal solar panels

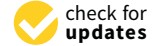



## 1. Introduction

Mandelbrot conceived the term 'Fractal' (in 1975) from the Latin word *fractus*, which means "broken" or "fractured" to describe irregular geometries in mathematics and in nature. Fractals are geometric objets that may repeat their geometry at smaller (or larger) scales due to the inherent self-similarity present in the object. Among several examples of well-known fractals, some classical examples are the Cantor set, the Sierpinski triangle, the Koch curve, the Mandelbrot set and Julia sets.

Many natural and man-made objects can be characterized using the classical Euclidean geometry and have integer dimension. However, the random geometry of natural objects such as a fern leaf, branching in human lungs, flowering head of broccoli, lightening during a storm, turbulence in a terrestrial body, coastlines, etc. can only be described more precisely using fractal geometry, and they have a non-integer fractal dimension.

Several hundred research articles are available in the literature covering various aspects of fractals including their mathematical development, scientific importance, engineering and industry applications. However, only a few references exists that cover a broader spectrum of fractals in one place, and most of these are in the form of monographs. Our prime objective of this survey is to provide a unified review of the work completed (over the past 5 decades) in the ever-growing field of fractal geometry covering length and breadth at once that will assist readers from various fields of academic and industry.

This comprehensive survey is written with the intent of providing a collative review of recent research, developments, and applications of fractals in a series of two papers. In Part-I [1], we covered a brief mathematical description of fractals, fractal dimension (which is usually a non-integer characteristic number attached to every fractal in contrast with Euclidean dimension) and applications of fractals in arts, tessellations, fashion designing, and other emerging fields such as econophysics, etc. This article is the second and last part of this survey with the aim of exploring engineering, industrial and commercial applications (including recent developments) of fractals in fractal landscapes, fractal antennas, fractal image compression, fracture mechanics and other evolving future applications of scientific and engineering research. We will see several fractal innovations which are making a great impact in modern technologies and will remain open for explorations in the future as well.

The article starts with an introduction to one of the most amazing discoveries in mathematics, namely the Mandelbrot set in Section 2. The space of fractals (the mathematical set where fractals live) and other elementary concepts are introduced in brief to give a flavor of the mathematics behind fractals to the reader, although the article is easy to follow by the majority of the scientific community without deeper understanding of mathematics.

Fractals are widely used for rendering landscapes in the computer graphics industry. The advent of fractal landscapes in computer graphics goes to Mandelbrot, who was the first to identify the similarity between the trace of fractional Brownian motion over time and the skyline of jagged mountain peaks [2] and explained the connection between visual approximation of natural mountains with the two-dimensional Brownian surfaces. This approach was implemented by Mandelbrot in [3] with the earliest computer graphics images of such surfaces and for the creation of fractal coastlines. Natural landscapes contain fractal characteristics and statistically self-similarity or self-affinity. In Section 3, we consider fractal landscapes, and standard algorithms for generating fractal landscapes are discussed.

In today's technology-driven world, antennas form an indispensable part of our life. They are used in cell phones, TV, radio, radars, WI-FI, IOT, bluetooth devices, and so on. There has been an incredible demand for the design of antennas that are compact and multiband or broadband. Properties of fractals can be exploited to achieve these multiple characteristics in a single antenna. Traditional antenna designs are based on Euclidean geometries; however, innovative antenna designs have emerged by exploiting the inherent self-similarity and space-filling properties of fractals. A fractal antenna is a revolutionary innovation in telecommunications. Fractal-shaped antennas have a large effective length, small size, and reduced weight with performance parameters, owing to the special geometry and compact structure of fractal shapes. Section 4 gives a detailed survey of different types of existing fractal antenna introduced over the last 2–3 decades along with historical developments and their applications in various communication systems.

Another important application of fractals is found in compressing data (e.g., images, music, and videos). Images are stored as a collection of bits representing pixels on a computer, and storing a collection of images requires large memory. This problem can be addressed using various image compression techniques that exist. Fractal Image Compression (FIC) is a powerful and evolving image compression technique, which is based on fractal coding that exploits the self-similarity property of an image. FIC is simple to implement, provides high compression ratios and fast decompression with the only drawback of slow compression. Barnsley introduced the fractal image compression in 1987, who founded Iterated Systems Inc. (a pioneer company in fractal image compression technology). In Section 5, we discuss various aspects, algorithms and applications of fractal compression.

Fracture mechanics is the study of propagation of cracks or failures of the structures in materials, and it is an important tool to improve the performance and quality of mechanical components. Mandelbrot was the first to interrelate the crack propagation and other fracture properties with the fractal geometry. He introduced the method of slit island analysis on the fracture surface to find fracture dimensions. Characteristics of the fractal geometry such as self-similarity (or self-affinity), scale invariance and fractal dimension

have offered great help to analyze irregular or fractional shapes of fracture mechanics. Section 6 discusses these aspects in more details.

Finally, in Section 7, biological applications of fractals are discussed with particular emphasis on ophthalmology. Other emerging applications of fractals such as fractal batteries, fractal electromagnets, fractal cooling chips, fractal PCBs, fractal solar panels, fractal capacitors, and fractals in biometric applications are also given here.

The two-part survey is organized is such a way that a reader will enjoy reading both parts independently without losing continuity and it will delight the readers with the applications of fractals in emerging and innovative fields of current and future technologies.

## 2. Mathematics of Fractals

Figure 1 shows Benoît Mandelbrot's eponymous set, which is popularly known as the **Mandelbrot set**, which is a mathematical fractal. The Mandelbrot set is among the most complex sets in mathematics and the best-known examples of mathematical visualization, self-similarities, and delightful patterns that are visible when we zoom on the set.

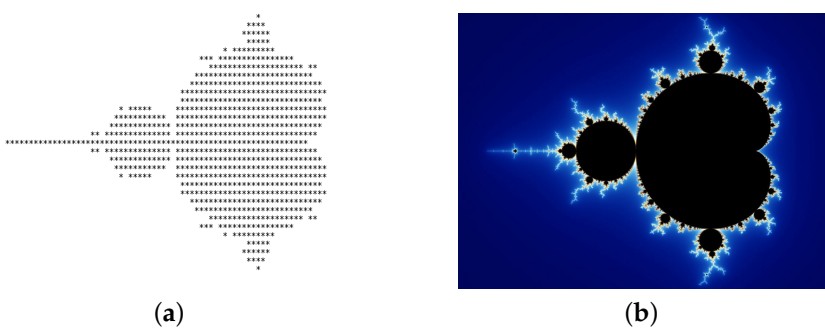

(a)  (b)

**Figure 1.** The Mandelbrot set: (**a**) first image (1978) and (**b**) image generated by Mandelbrot (1980).

R. Brooks and P. Matelski published the first image (Figure 1a) of the Mandelbrot set in the year 1978. Later, Mandelbrot plotted the true image of the Mandelbrot set on 1 March 1980 (Figure 1b). This set is obtained by plotting the complex numbers $c$ in the simple (quadratic) polynomial

$$f_c(z) = z^2 + c,$$

whose orbits remain bounded. Generalized Mandelbrot sets can also be plotted by considering the higher degree polynomials

$$f_c(z) = z^n + c, \quad n > 1.$$

In Figure 2a–d, generalized Mandelbrot sets are displayed for $n = 3, 4, 5$ and 10. We refer to [4] for an interesting work on generalized Mandelbrot sets with chaotic features obtained by replacing $z^2$ with Möbius transformations, transcendental functions, etc. Some properties of these generalized sets are also discussed in contrast with the original Mandelbrot set [4].

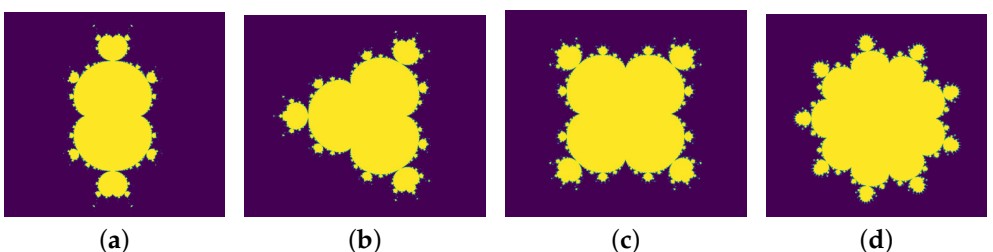

(a)  (b)  (c)  (d)

**Figure 2.** Generalized Mandelbrot Sets for (**a**) $n = 3$ (**b**) $n = 4$, (**c**) $n = 5$ and (**d**) $n = 10$.

The Mandelbrot set has become so popular that this set and its details (the Julia sets which live on the boundary of the Mandelbrot set) can be seen on cloths, ceramic products, tiles, hot air balloons, calenders, art prints, postcards, posters, commercials and so on. For an incredible zoom on the Mandelbrot set, we refer to [5].

### 2.1. Space of Fractals

Let $X$ be a non-empty set, a function $d : X \times X \to \mathbb{R}^+$ is called a metric or a distance function on $X$ if it satisfies

$$
\begin{array}{lll}
(i) & d(x,y) \geq 0 \quad \text{and} \quad d(x,y) = 0 \Leftrightarrow x = y, & \forall \quad x, y \in X, \\
(ii) & d(x,y) = d(y,x), \quad \forall \quad x, y \in X, \\
(iii) & d(x,y) \leq d(x,z) + d(z,y), \quad \forall \quad x, y, z \in X.
\end{array}
$$

The set $X$ together with the function $d$ is called a metric space, and it is denoted by $(X, d)$.

A metric space $X$ is said to be *complete* if every Cauchy sequence is convergent in $X$ and a subset $S \subseteq X$ is said to be *compact* if every infinite sequence of points in $S$ has a convergence subsequence. A complete metric space and its compact subsets are fundamental tools to describe and understand fractal geometry, which is essentially the classification, description, analysis and observations of subsets of metric spaces.

**Definition 1.** *Let $(X, d)$ be a complete metric space and $\mathcal{H}(X)$ be the set of non-empty compact subsets of X. For any $A, B \in \mathcal{H}(X)$, define the distance between A and B by*

$$
h(A, B) = max\{d(A, B), d(B, A)\},
$$

*where $d(A, B) = sup_{x \in A} \, inf_{y \in B}\{d(x, y)\}$.*

Then, it is easy to verify that $h$ is a metric on $\mathcal{H}(X)$. This metric $h$ is called the Hausdorff metric on $\mathcal{H}(X)$, and the set $\mathcal{H}(X)$ is called the **space of fractals** equipped with the Hausdorff metric $h$.

**Theorem 1.** *The space $(\mathcal{H}(X), h)$ is a complete metric space.*

**Proof.** See Barnsley [6] (Chapter 2). □

Any subset of $(\mathcal{H}(X), h)$ is a mathematical fractal, although the Euclidean objects such as rectangles, parallelograms, spheres and cylinders are not considered as fractals, since they do not possesses self-similarity, but they are elements of $(\mathcal{H}(X), h)$ and can be considered as (mathematical) fractals if there no confusion is likely to occur.

### 2.2. Iterated Function Systems and Attractors

**Definition 2.** *A mapping or a transformation $w : X \to X$ on a metric space $(X, d)$ is called a contraction mapping if*

$$
d(w(x), w(y)) \leq \alpha \, d(x, y) \quad \forall x, y \in X. \tag{1}
$$

*for some constant $0 \leq \alpha < 1$. The constant $\alpha$ is called contractivity factor of w.*

**Definition 3.** *A finite set of contraction mappings $w_i : X \to X$, where X is a metric space equipped with the metric d having contractivity factors $\alpha_i$, for $i = 1, 2, \ldots, m$ is called an iterated function system (IFS). The number*

$$
\alpha = \max_{1 \leq i \leq m} \alpha_i,
$$

*is called a contractivity factor of the IFS.*

**Theorem 2** (Hutchinson [7]). *Let $\{X, w_i : i = 1, 2, \ldots, m\}$ be an IFS with contractivity factor $\alpha$. Then, the transformation $W : \mathcal{H}(X) \to \mathcal{H}(X)$ defined by*

$$W(B) = \bigcup_{i=1}^{m} w_i(B),  \qquad (2)$$

*for all $B \in \mathcal{H}(X)$ is a contraction mapping on $\mathcal{H}(X, h(d))$ with contractivity factor $\alpha$.*

Therefore, by the contraction mapping theorem, the mapping $W$ has a unique fixed point $A \in \mathcal{H}(X)$ given by

$$A = \lim_{n \to \infty} W^{\circ n}(B), \quad B \in \mathcal{H}(X).$$

Here, $W^{\circ m}(B)$ denotes the $m$-fold forward iterate of $W$.

**Definition 4.** *The unique fixed point $A$ described in Theorem 2 is called the **attractor** of the IFS. Moreover, since $A \in \mathcal{H}(X)$, therefore, it is a (mathematical) fractal.*

The examples of mathematical and natural fractals to be presented in the ensuing sections of this article are geometrically intricate subsets of Euclidean spaces $\mathbb{R}^2$ or $\mathbb{R}^3$, which are elements of $\mathcal{H}(X)$ with $X = \mathbb{R}^d, d = 2, 3$.

## 3. Fractals in Natural and Artificial Landscapes

A fractal landscape is typically a surface that displays fractal behavior obtained by an algorithm and mimics the appearance of a natural terrain. Midpoint displacement methods by Fournier et al. [8], Miller [9], Musgrave [10] and others were introduced as fast landscape and terrain generation techniques and are standard in fractal geometry. Ken Musgrave (a student of Mandelbrot) discovered new processes of fractal landscape generation [10]. He worked on Bryce landscape software, which made use of many algorithms (midpoint displacement algorithm was one of those). The midpoint displacement methods were modified and improved in [11,12] for natural terrain simulations and to construct self-affine geometrical objects which are similar to rock fractures.

Examples of natural fractal landscapes are found in geography, mountains, rivers, and terrains. A natural fractal mountain is shown in Figure 3, and a natural delta formed by a flowing river and a fractal shape profile of clouds is displayed in Figure 4.

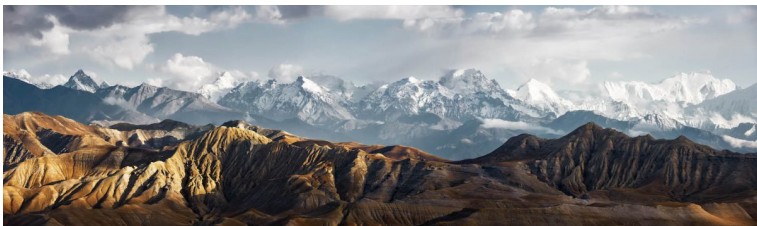

**Figure 3.** A fractal mountain.

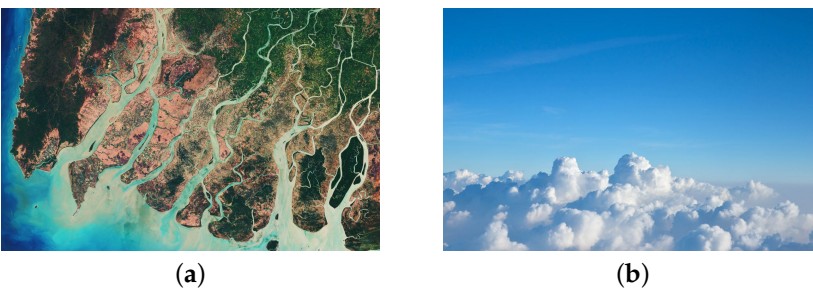

|(a)|(b)|
|---|---|

**Figure 4.** (**a**) A fractal river detla, (**b**) a fractal sky cloud.

F. Kenton Musgrave was the first to generate computer-based realistic landscapes. He was referred to as "*the first true fractal-based artist*" by Mandelbrot for his Ph.D. thesis work on *Methods for Realistic Landscape Imaging* [10]. Musgrave's thesis work turned out to be a comprehensive road map for rendering modern fractal landscapes using computer programs even today. Musgrave founded the Pandromeda, Inc. and developed the innovative MojoWorld software (obsolete now), a commercial and fractal-based modeling program for the creating digital landscapes, space art and science fiction scenes. The MojoWorld was applied in creating background mattes and terrains on big-budget movies such as *Titanic*, *The Day After Tomorrow*, etc. Figure 5 shows realistic examples of computer-generated fractal landscapes. Notice the true similarities between Figures 3 and 5.

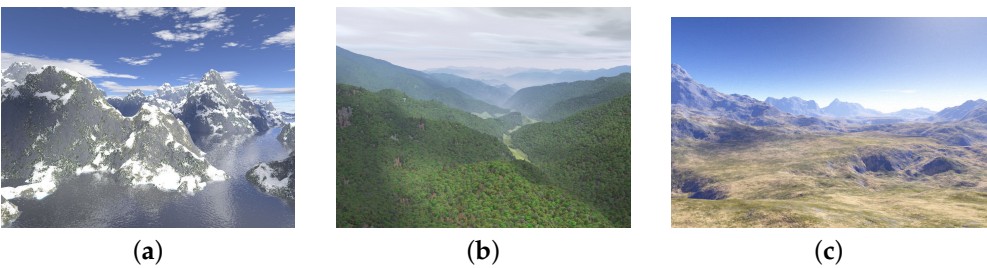

(a)          (b)          (c)

**Figure 5.** Computer-generated examples of (**a**) a fractal terrain, (**b**) a fractal woodhill, and (**c**) a fractal landscape. (Image source: https://en.wikipedia.org/wiki/Fractal_landscape, accessed on 22 June 2022).

### 3.1. Generation of Fractal Landscapes

There is a large variety of commercial and academic purpose software that can generate and allow for editing of fractal landscapes. The list includes Bryce (a feature-packed 3D modeling and animation package specializing in fractal landscapes), midpoint displacement algorithm (landscapes generation in many dimensions), diamond-square algorithm [8] (slightly better algorithm than midpoint displacement algorithm), Terragen (designed and developed by the Planetside Software for Microsoft Windows and Mac OS X and capable of generating captivating sceneries and animations of fractal landscapes), L3DT (similar as the Terragen program with a 2048 × 2048 limit) and World Creator (can create terrain, fully GPU powered), etc. Figure 6a displays a Julia island, and an example of a *Mandel River* generated by the software Terragen is shown in Figure 6b, which depicts the details of the Mandelbrot set.

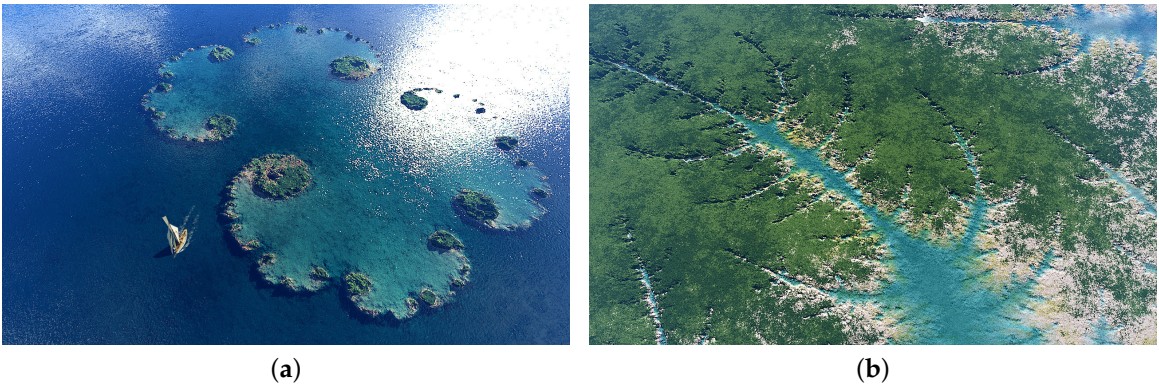

(a)                    (b)

**Figure 6.** (**a**) Julia island (Image source: https://en.wikipedia.org/wiki/Terragen, accessed on 22 June 2022) and (**b**) Mandel river (details of the Mandelbrot set) rendered by Terragen Classic. (Image source: https://en.wikipedia.org/wiki/Fractal-generating_software, accessed on 22 June 2022).

We now describe some of the above-mentioned fractal rendering algorithms to allow the reader deeper insight and better understanding on how the fractal landscape generation algorithms work.

### 3.2. Midpoint Displacement Algorithm in 1d (1DMD)

The midpoint displacement algorithm is based on the von Koch curve construction. The credit for its applicability and popularity in computer graphics goes to Fournier, Fussell, and Carpenter for rendering fractal landscapes and clouds. The algorithm is very simple and proceeds as follows:

Start with a straight line segment and mark its midpoint. Now, select a random (bounded) value and displace the midpoint of the line segment by this random value in the direction perpendicular to the line segment or displace only the $y$ coordinate of the midpoint (see Figure 7a).

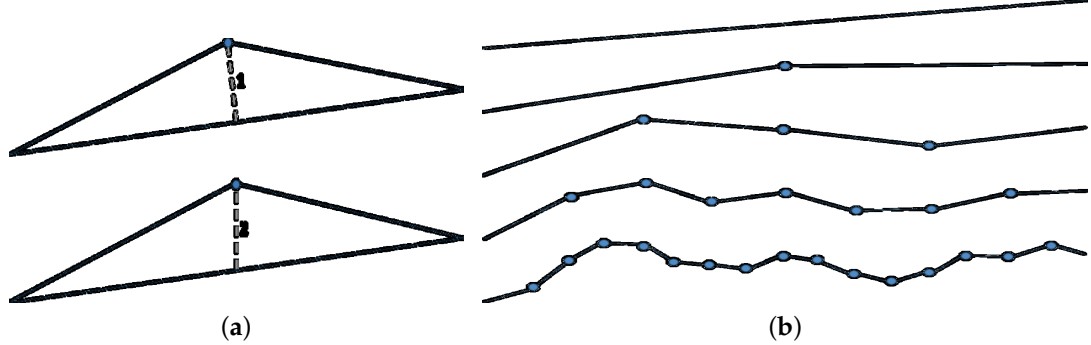

| (a) | (b) |

**Figure 7.** (**a**) Strategies to displace the midpoint and (**b**) Successive iterations of the algorithm (from left to right). (Image source: https://bitesofcode.wordpress.com/2016/12/23/landscape-generation-using-midpoint-displacement/, accessed on 22 June 2022).

This will result in two smaller line segments. In the second iteration, repeat this process to mark and displace the midpoints of each line segment obtained in the first iteration by a random amount, and this will result in four straight line segments. The process is continued until the desired level of detail is achieved by reducing the random displacement in every iteration. For example, if the displacement was reduced by half in the first iteration and the random displacement value is chosen from the interval $[-1, 1]$, then the range for the second iteration with two midpoints would be in the interval $[-0.5, 0.5]$, in $[-0.25, 0.25]$ for the third iteration, and so on. The equation for the midpoint value is given by

$$F(x) = \frac{\left(F\left(x + \frac{dx}{2}\right) + F\left(x - \frac{dx}{2}\right)\right)}{2} + Kr \cdot 2^{-nH}, \tag{3}$$

where $r \in [-1, 1]$ is a random number and $K$ is a constant which controls the amplitude of the variation. $H$ is the roughness parameter (the factor by which the perturbations are reduced on each iteration), and $n$ denotes the iteration number. Increasing the value of $H$ will produce smoother landscapes. Figure 7b displays successive iterations of the algorithm. The pseudocode for the algorithm is given in Algorithm 1.

By suitably choosing the displacement bounds and the reduction factor $H$, one can control the geometry and the roughness of the landscape. Higher values of $H$ result in smoother landscapes, and lower values result in spiky (sharp) landscapes. Figure 8 depicts several landscapes with varying $H$ values. Observe the change in the smoothness of the landscape with the change in $H$ values.

**Algorithm 1:** Pseudocode for midpoint displacement algorithm.

**Pseudocode:**
initialize *line segment*
initialize *max_iter*, *min_len*
while *iteration < max_iter* and *segment_length > min_len*:
    for each segment:
        *choose random displacement*
        *compute midpoint*
        *displace midpoint*
        *update segments*
    *reduce displacement*
    *iteration+1*

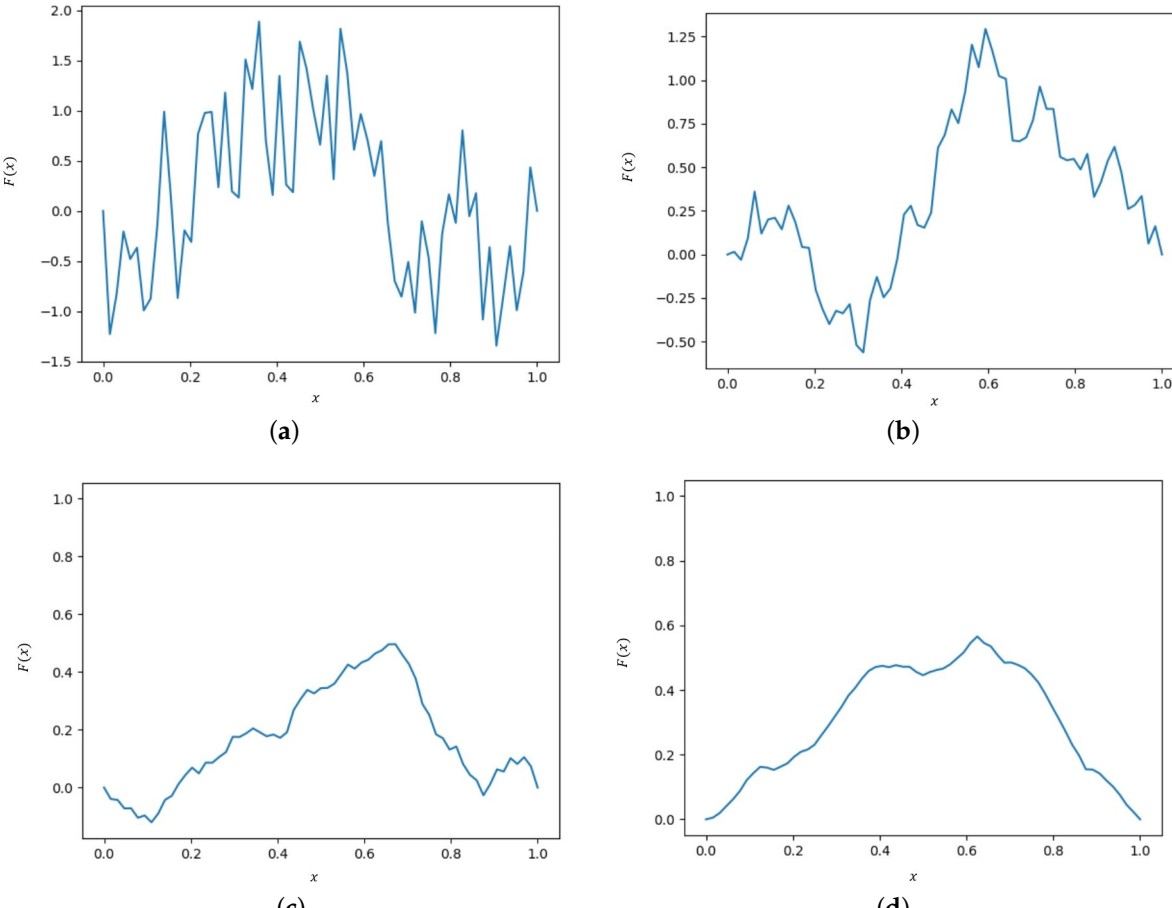

**Figure 8.** One-dimensional (1D) landscapes for (**a**) $H = 0.0$, (**b**) $H = 0.25$, (**c**) $H = 0.50$, and (**d**) $H = 0.75$.

In each iteration, the displacement bounds can be reduced by different approaches (e.g., linear, exponential, logarithmic, etc.) depending upon the choice of landscape being generated. The two extremes possibilities are no displacement reduction and exponential displacement reduction (in each iteration) shown in Figure 9 below.

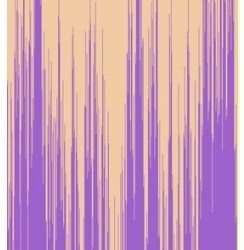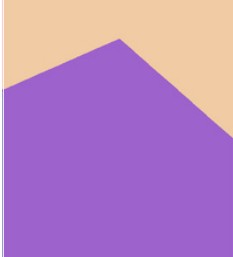

**Figure 9.** No displacement reduction (**left** image), Exponential displacement reduction in one iteration (**right** image). (Image source: https://bitesofcode.wordpress.com/2016/12/23/landscape-generation-using-midpoint-displacement/, accessed on 22 June 202).

Some colored pictures of landscapes generated from the 1D midpoint displacement algorithm are presented in Figure 10.

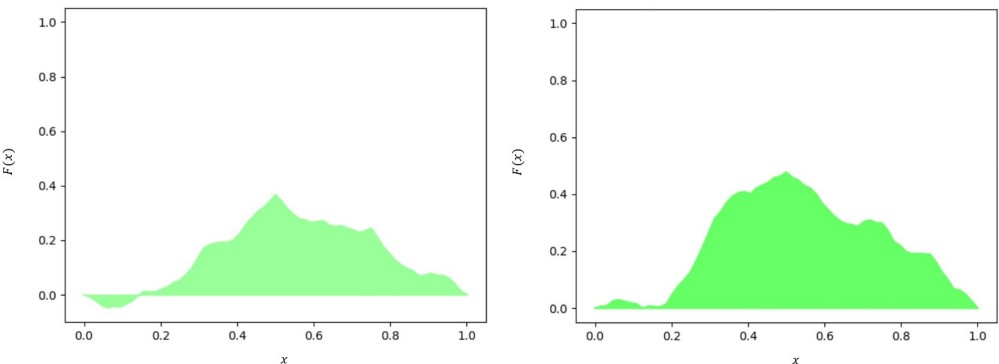

**Figure 10.** Colored landscapes generated from 1DMD.

### 3.3. Midpoint Displacement Algorithm in 2D (2DMD)

The 2D midpoint displacement algorithm is similar to the 1D algorithm described above, with the only difference that now, the displacement (height) in the $z$-direction is determined over the $xy$-plane. In most cases, a positive displacement results in the formation of a mountain, and a negative displacement results in the formation of a valley. The advantage of using this algorithm is that the landscapes are dynamically generated, and they will never be the same, as the elevations chosen are random every time.

The roughness of the landscape is controlled in the same way as in a one-dimensional landscape. Changes in $H$ values show drastic changes in the landscape generated: for instance, if the value of $H$ is 0, then the landscape is more spiky, and when it is 1, we obtain smooth landscapes, as seen in surface landscapes generated using 2DMD in Figure 11.

An extension of the 2D midpoint displacement algorithm to three-dimensions was presented in [13] for generating three-dimensional fractal porous media geometries whose surface area can also be controlled by adjusting the random component of the midpoint displacement. They also considered statistical properties for the geometries obtained using 3DMD and showed that the structures generated by 3DMD are more realistic.

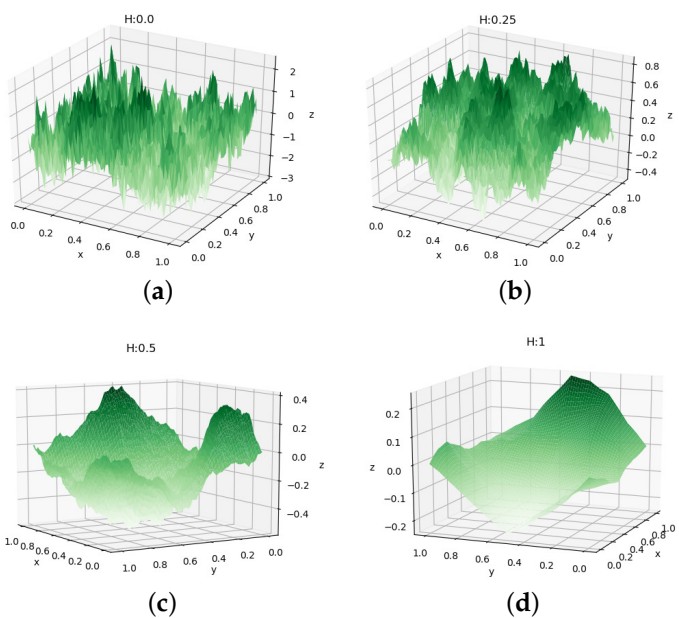

**Figure 11.** Surface landscapes generated by 2DMD for (**a**) $H = 0.0$, (**b**) $H = 0.25$, (**c**) $H = 0.50$, and (**d**) $H = 1.0$.

The grids used with the midpoint displacement algorithm are uniform in all directions, and typically, they have a size of $2^n$ on all sides, where $n$ is an integer. The variable $n$ (number of iterations) which is given as an input (by the user) has its own significance. Increasing the value of $n$ leads to an increase in the resolution of the landscape, as minute details of fractals will be captured. However, generating fractals with high values of $n$ is a time-consuming process and requires high computational powers, so it is important to select an optimal value of $n$ by taking into consideration the time, computational power and required resolution.

*3.4. Diamond Square Algorithm*

The diamond square algorithm is a modification of the midpoint displacement method proposed by Fournier et al. [8] (1982), and its name is borrowed from the 2D midpoint displacement algorithm. The midpoint displacement method sometimes leaves square-shaped artifacts in generated terrains. The diamond square algorithm attempts to alleviate this by alternating calculated values to square and diamond patterned midpoints. The algorithm starts with a $2D$ square grid of boxes having $2^n$ squares containing $2^n + 1$ grid points. The four corner points of the grid are first set to initial values. The diamond and square steps are then executed one after the other until all grid points have been assigned as follows:

- *The diamond step*: For each square in the array, set the midpoint of that square to be the average of the four corner points plus a random value.
- *The square step*: For each diamond in the array, set the midpoint of that diamond to be the average of the four corner points plus a random value.

Figure 12 shows the algorithmic steps of the algorithm. The magnitude of the random value should be reduced in each iteration.

Miller [9] analyzed the diamond square algorithm in 1986 and described it as flawed due to possible perturbations in the rectangular grid. The grid artifacts were resolved by J.P. Lewis in a generalized algorithm [14]. Some landscapes images generated by the diamond square algorithm at different $H$-values are shown in Figure 13.

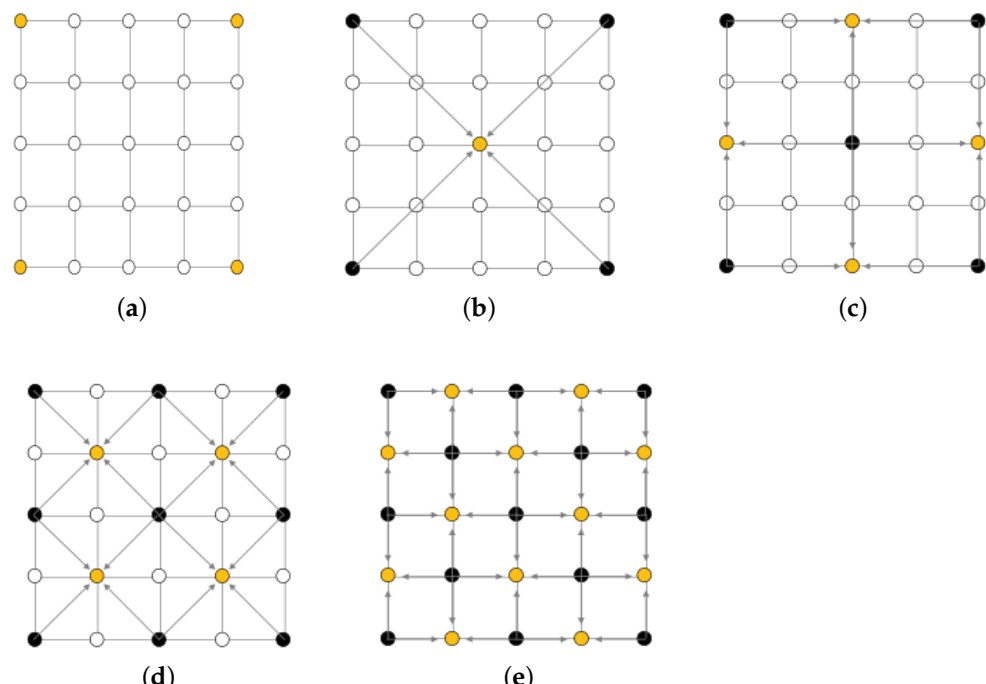

**Figure 12.** Diamond square algorithm on a 5 × 5 array: (**a**) Initialize corner grid values, (**b**) execute diamond step, (**c**) execute square step, (**d**) execute diamond step, (**e**) execute square step. (Image source: https://en.wikipedia.org/wiki/Diamond-square_algorithm, accessed on 22 June 2022).

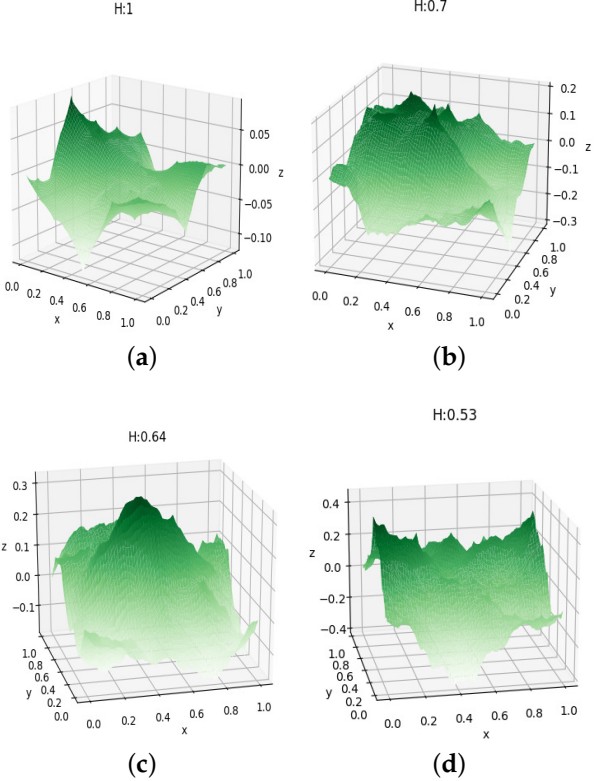

**Figure 13.** Surface landscapes from diamond square algorithm at different values of *H*: (**a**) *H* = 1.0, (**b**) *H* = 0.70, (**c**) *H* = 0.64, and (**d**) *H* = 0.53.

### 3.5. Summary

According to Musgrave [10], the generation of realistic fractal landscapes or creating *fractal forgeries of nature* consists of geometric models, designing efficient algorithms, atmospheric effects (for sense of scale), surface textures, and a global context for embedding the scenes. In this brief essay on fractal landscapes, we briefed the pioneering work by several authors including the work of Musgrave on analysis and algorithms that are available for creating fractal landscapes. The review highlights the potential of fractal geometry to understand and design fractal landscapes. Fractal landscape generation is evolving rapidly, and the design of new and fast algorithms is still under development.

## 4. Fractal Antennas

Antennas are an integral part of any communication system, and they are widely used in electromagentic devices such as cell phones, TV, radio, radars, electronic devices, and so on. With the advancement of technology, the world is becoming more dependent on compact, bluetooth, WI-FI and IOT smart devices. Therefore, the need is to design antennas for commercial and defence sectors that are compact, light weight, and multiband or broadband. A natural choice to obtain these antenna characteristics is to exploit the properties of fractals. Today, many novel and powerful antenna designs have emerged from modern (fractal) geometry, which are replacing the traditional antenna designs based on Euclidean geometries.

A fractal antenna is a revolutionary invention in the field of telecommunication. Using a fractal-shaped antenna as a replacement of a circuit with discrete components has helped in increasing the effective length and reducing the size and weight of the antenna. At the same time, the performance parameters have improved, owing to the self-similar geometry (which maximizes the effective length of an antenna for a given surface area) and compact structure of fractal shapes. A large number of fractal antenna designs have been proposed combining fractal geometry with electromagnetic theory, and this has led to an area called fractal antenna engineering [15].

In this section, we review standard fractal-shaped antennas proposed and simulated by many researchers in the past two decades, since the pioneering works of Cohen and Puente [16–19]. The work by Werner et al. [15] summarizes various techniques for compact (i.e., miniature) fractal antennas designs. We also refer to the recent survey papers [20,21] for an extensive study of the literature and state of the art summary of fractal antenna research. The reader may also consider exploring the articles [19,20,22,23] for more detailed analysis, various types and applications of fractal antennas available in the literature.

### 4.1. Brief History

Nathan Cohen was the first to built a wire fractal antenna using von Koch curves in 1988 (at Boston University) by setting up a 'ham' radio station, and he also designed the planar fractal arrays using Sierpinski triangles. Cohen co-founded Fractal Antenna Systems Inc. in 1995 as the first fractal-based commercial antenna solutions, and he also designed fractal cellular antennas for Motorola phones, which were proven to be 25% more efficient than the conventional helical antenna. Another company founded by C. Puente and R. Bonet, namely Fractus S.A. in Barcelona (Spain), is involved in fractal antenna research, patents and commercialization.

In August 1995, Cohen published the first article on fractal antenna [16], and Puente carried out early work on fractals as multiband antennas [24]. Therefore, the credit for demonstrating the potential of fractal antennas as a replacement for traditional antennas is jointly shared by Cohen and Puente.

Because of their special geometry, fractal antennas are self-loading and often do not need matching circuitry for multiband or broadband characteristics. This lowers the fabrication cost and increases the reliability. Exploiting the self-similar fractal designs, one can fabricate fractal antennas that are compact and wideband. The fractal-shaped antennas can have multiple resonances (self-similar design works as a virtual network of

capacitors and inductors), making a single antenna operate on multiple electromagnetic frequencies. Due to space-filling properties, fractal antennas make better use of the available volume inside the radian sphere. Therefore, they may radiate more effectively than the one-dimensional straight wire [18].

*4.2. Antenna Parameters*

While designing an antenna, one must consider different combinations of antenna parameters based on the type of application for which the antenna is being fabricated. For instance, antennas used for television must have higher bandwidths to support higher data transmission rates. For radio, the antennas' range and capability to work at multiple bands is considered more important, and for modern antennas, the size of the antenna matters a lot, since nanotechnology is the direction in which the world is moving. Thus, antenna parameters play a vital role in the design, fabrications and applications. Before we look at some examples of fractal antennas, let us briefly describe some of the key antenna parameters.

4.2.1. Impedance

Transmission lines are used to feed antennas, and to transmit the maximum available power or to receive the transmitted power, it is necessary to know the impedance at the input where the transmission line is to be connected. For optimal power transfer from the antenna to the receiver or from the transmitter to the antenna, the input impedance of the transmission line must be same as the input impedance of the antenna. In case of impedance mismatch, an impedance matching circuit is required.

4.2.2. Return Loss

The return loss compares the power reflected by the antenna to the power that is fed into the antenna from the transmission line. It is measured in dB, and the relationship between SWR (Standing Wave Ratio, a measure of impedance matching) and return loss is given by

$$\text{Return loss}(dB) = 20 \, \log_{10} \frac{SWR}{SWR - 1}.$$

4.2.3. Bandwidth

Bandwidth refers to the range of frequencies over which the antenna can properly radiate or receive energy. The desired bandwidth is one of the key parameters for an antenna design. The antenna's bandwidth is the number of Hz for which the antenna will exhibit an SWR less than 2:1. The bandwitdh of an antenna is defined by

$$B = f_h - f_l,$$

where $B$ =Bandwidth, $f_h$ =Higher cut-off frequency, $f_l$ =Lower cut-off frequency.
The bandwidth can also be described in terms of percentage of the center frequency of the band

$$B = \frac{f_h - f_l}{f_c} \times 100,$$

where $f_c$ is the center frequency in the band. Bandwidth is typically quoted in terms of VSWR. The bandwidth of an antenna varies according to its type and application.

#### 4.2.4. Directivity

Directivity is the ability to focus the concentration of an antenna's radiation pattern in a particular direction when transmitting or to receive energy from a particular direction. Directivity is denoted by $D$ (expressed in dB) and defined by

$$D = \frac{F_{\text{max}}}{F_{\text{iso}}},$$

where $F_{\text{max}}$ =maximum signal strength radiated by the antenna, $F_{\text{iso}}$ =maximum signal strength radiated by the isotropic antenna (an antenna that radiates power equally in all directions).

#### 4.2.5. Antenna Efficiency

The efficiency of an antenna is the ratio of the power radiated by the antenna to the power radiated from the antenna.

$$\eta = \frac{P_{radiated}}{P_{input}},$$

where $\eta$ =antenna efficiency, $P_{radiated}$ =power radiated, and $P_{input}$ =input power to the antenna.

#### 4.2.6. Antenna Gain

The term antenna gain describes how much power is transmitted in the direction of peak radiation to that of an isotropic source. An antenna's gain ($G$) is a key parameter that combines an antenna's radiation efficiency ($\eta$) and directivity ($D$) by the relation:

$$G = \eta \times D.$$

The antenna gain is expressed in decibels (dB) by:

$$G_{dB} = 10 \cdot \log_{10}(G)$$

In principle, a high-gain antenna will radiate most of its power in one direction, and a low-gain antenna will radiate its power equally in all directions.

#### 4.2.7. Radiation Pattern

The radiation pattern displays the variation of the power radiated by an antenna as a function of the direction away from the antenna. That is, the antenna's pattern describes how the antenna radiates energy out into space (or how it receives energy). A radiation pattern is "isotropic" if the radiation pattern is the same in all directions. Antennas with isotropic radiation patterns do not exist in practice, but they are used for benchmarking with real antennas.

#### 4.2.8. Polarization

The polarization of an antenna is defined as the direction of the electromagnetic fields produced by the antenna as energy radiates away from it, with respect to the surface of the earth, and it is determined by the structure of the antenna and its orientation. These directional fields determine the direction in which the energy moves away from or is received by an antenna.

There are several categories of polarization, and within each type, there are several sub categories such as linear polarization (horizontal, vertical and slant), circular polarization (right-hand circular and left-hand circular), elliptical polarization, omnidirectional polarization, etc.

4.2.9. Types of Antennas

Antennas are classified in many categories based on their physical structure, functionality and types of applications. Well-known examples of antennas including their types and application areas are

1.  Wire antennas (e.g., dipole antenna, monopole antenna, helix antenna, loop antenna), used in personal applications, buildings, ships, automobiles, space crafts, etc.
2.  Aperture antennas (e.g., waveguide (opening), Horn antenna), used in flush-mounted applications, aircrafts, spacecrafts, etc.
3.  Reflector antennas (e.g., parabolic reflectors, corner reflectors) used in microwave communication, satellite tracking, radio astronomy.
4.  Lens antennas (e.g., convex–plane, concave–plane, convex–convex, concave–concave lenses), used for very high-frequency applications.
5.  Microstrip antennas (e.g., circular-shaped, rectangular-shaped metallic patch above the ground plane), used in aircrafts, spacecrafts, satellites, missiles, cars, mobile phones, etc.
6.  Array antennas (e.g., Yagi-Uda antenna, microstrip patch array, aperture array, slotted wave guide array), used for very high-gain applications.

4.2.10. Substrate

Low-profile antennas are needed for high-performance aircrafts, spacecrafts, satellites, missile applications, GSM, GPS and remote sensing applications where size, performance, weight, cost, ease of installation, and aerodynamic profile are constraints. All these requirements may be met using a microstrip antenna (MSA). An MSA (also called patch antenna) is a two-dimensional flat structure consisting of a very thin metallic strip placed on a ground plane with a dielectric material in between; this dielectric material is called the *substrate*.

The performance and radiation properties of an antenna can be improved by properly selecting the thickness ($h$) and permittivity ($\epsilon_r$) of the substrate. In patch antennas, the smaller permittivity of the substrate yields better radiation. Several dielectric substrates are proposed in the literature for fabricating microstrip patch antennas. Table 1 lists some commonly used substrate materials in the design of fractal antennas along with their dielectric constants.

**Table 1.** Commonly used substrate materials in fractal antennas.

| S. No. | Name of the Substrate | Dielectric Constant ($\epsilon_r$) |
| :---: | :---: | :---: |
| 1. | Bakelite | 4.8 |
| 2. | Duroid 6010 | 10.7 |
| 3. | Nylon fabric | 3.6 |
| 4. | Roger 4350 | 3.48 |
| 5. | RT-Duroid | 2.2 |
| 6. | Foam | 1.05 |
| 7. | Taconic TLC | 3.2 |
| 8. | FR-4 | 4.4 |

*4.3. Standard Fractal Antennas*

The first application of fractal antennas was in the form of wire antennas proposed by Cohen in a series of papers [16,17] based on fractalization of the geometry of a standard dipole or loop antenna. Almost at the same time, Puente and his collaborators [18,19,24] proposed Koch fractal monopole antennas with improved electrical performance over conventional linear monopole antennas.

Cohen observed that fractal Minkowski loops exhibit low resonant frequency relative to their electric size. Puente found that in Koch fractal antennas, the resonant frequency goes low or toward larger wavelengths with increase in iteration number. Thus, fractal-shaped antennas at higher iterations resonate at low frequencies due to increased length as compared to the antennas of the lower iterations (having smaller length).

Today, most of the wireless devices operate in multiple bands of frequencies. Thus, the design of a multiband antenna is a natural choice for present and future devices. We now provide a brief overview of some popular fractal-shaped antennas, which are proven to be very useful in developing novel, innovative designs for multiband fractal antennas. To keep the presentation shorter, we provide plots for multiband behavior only for the Sierpinski gasket antenna, and we encourage the reader to consider references mentioned here for details on the design, performance and applications of fractal antennas.

### 4.3.1. Sierpinski Gasket

Figure 14 shows the first five stages in the construction of the Sierpinski gasket antenna (named after the Polish mathematician Sierpinski.

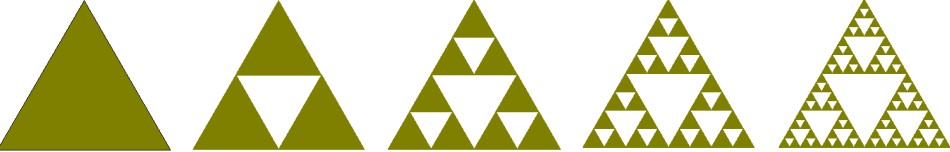

**Figure 14.** Sierpinski gasket antenna through five stages of growth.

The Sierpinski gasket is obtained by continuing the iterations to infinity. From an antenna engineering perspective, the colored (filled) triangular regions represent a metallic conductor, whereas the white (hollow) triangular regions represent areas where the metal has been removed. The self-similar geometry of Sierpinski gaskets allows for fabricating multiband fractal antenna elements.

The Sierpinski gasket antennas resemble a bow-tie antenna, and one antenna can perform similar to multiple bow-tie antennas, since the iterated Sierpinski gasket consists of many Sierpinski gaskets at different scales, which can be seen by looking at Figure 15a. A fabricated Sierpinski gasket antenna is shown in Figure 15b, and the lengths of the largest side of the antenna are shown in Figure 15c at various scales.

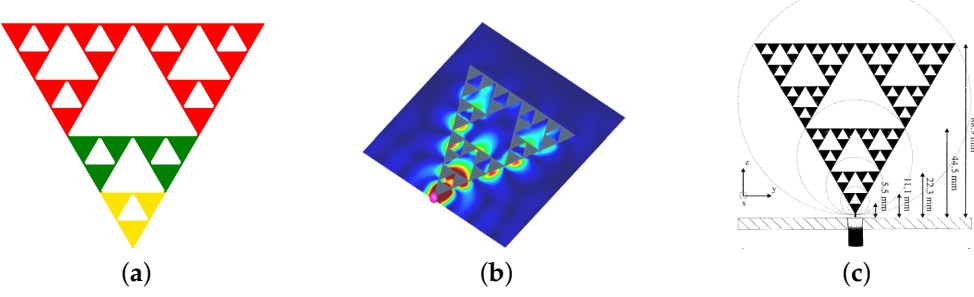

(**a**) (**b**) (**c**)

**Figure 15.** Resemblance of Sierpinski gasket antenna to bow-tie antenna. (**a**) Three stages of Sierpinski antenna, (**b**) Fabricated Sierpinski antenna, (**c**) Length scales of Sierpinski antenna. (Image source: https://www.emcos.com/wp-content/uploads/2014/01/Application_Note_Fractal_Antennas_Simulation_Sierpinski_Gasket.pdf, accessed on 22 June 2022).

The multiband performance of this Sierpinski antenna is visible in Figure 16, where some plots are given between the $S_{11}$ parameter (which gives the amount of power reflected from the antenna) and the frequency. $S_{11} = 0$ signifies that all the power is reflected, so the negative sharp down peaks are considered as the resonating frequencies.

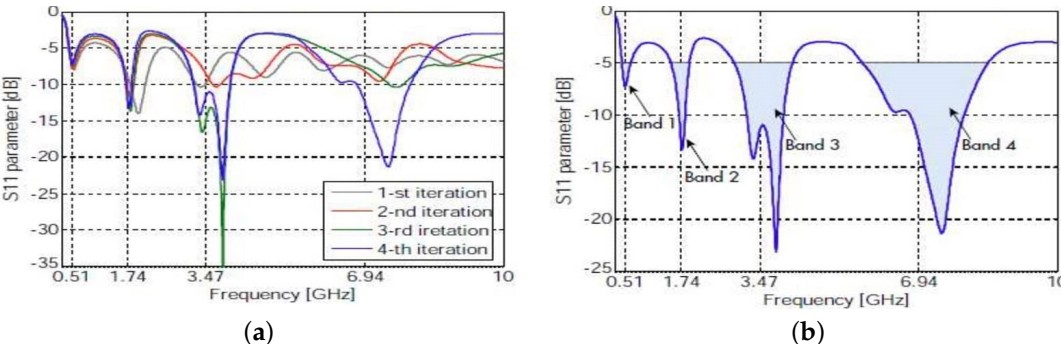

(a)                                    (b)

**Figure 16.** $S_{11}$ plots for Sierpinski gasket antenna (**a**) all iterations (**b**) 4th iteration. (Image source: https://www.emcos.com/wp-content/uploads/2014/01/Application_Note_Fractal_Antennas_Simulation_Sierpinski_Gasket.pdf, accessed on 22 June 2022).

The simulated characteristics of the Sierpinski gasket monopole antenna were shown to be matching with the analytical results in [25]. Moreover, the antenna resonates at multiple frequencies, making the Sierpinski gasket antenna a multiband antenna. Vertical and horizontal polarization plots for the 4th frequency band are shown in Figure 17.

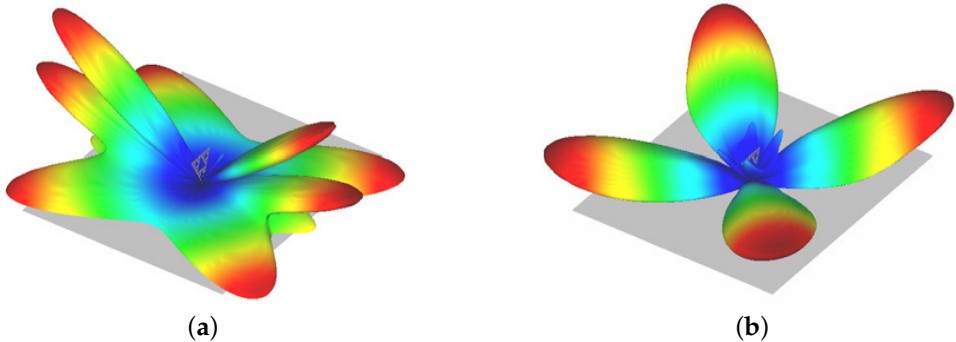

(a)                                    (b)

**Figure 17.** Fourth frequency band: (**a**) Vertical polarization and (**b**) Horizontal polarization. (Image source: https://www.emcos.com/wp-content/uploads/2014/01/Application_Note_Fractal_Antennas_Simulation_Sierpinski_Gasket.pdf, accessed on 22 June 2022).

The simulations and plots in Figures 16 and 17 are drawn using the EMCoS Antenna VLab environment, which is a software for electromagnetics, data visualization and simulation.

Simulations for other type of fractal antennas can be completed in a similar way using any EM simulation software (e.g., HFSS, CST Studio, EMCoS, COMSOL, etc.), and the details are available in many references cited throughout this section; therefore, we shall omit simulation details for other antennas to keep the presentation short.

4.3.2. Koch Curve

Figure 18 shows the first four iterations in the construction of the Koch curve monopole antenna, which became the first small size fractal antenna that improved bandwidth, resonance frequency, and radiation patterns of classical antennas in 1998. The von Koch curve is obtained by subdividing a line segment into three parts.

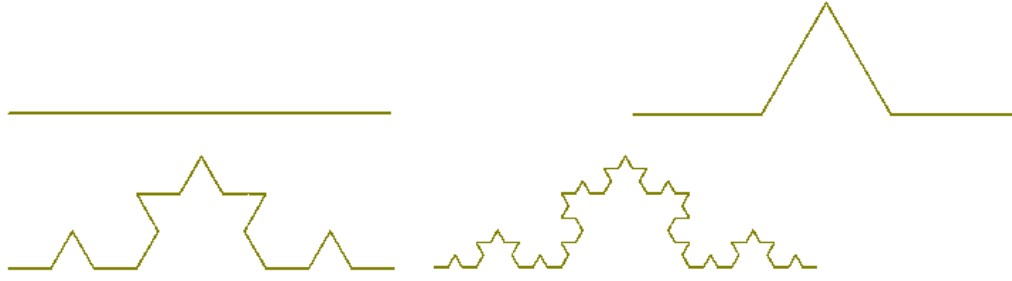

**Figure 18.** Four stages of Koch fractal.

The middle part is then replaced by adding two sides of an equilateral triangle having the length equal to the length of the segment being removed. This results in four line segments. Repeating this process for each of the four segments and taking the limit constitutes the Koch curve.

Puente et al. [19] studied the von Koch fractal as a monopole wire antenna. They considered five different iterations of the von Koch antenna, having an overall height $h = 6$ cm, and a total length of $L = h \times \left(\frac{4}{3}\right)^5 = 25.3$ cm (see [19] for complete analysis and simulation reults). In general, the length of the Koch curve can be determined by formula $L_n = \left(\frac{4}{3}\right)^n$ ($L_n$ is the length of the Koch curve at the $n$th iteration). Since $\frac{4}{3} > 1$, therefore, as $n \to \infty$, the length of the Koch curve will tend to infinity. So, theoretically, we can design an antenna of desired length in a given area using the Koch curve.

### 4.3.3. Koch Snowflake

Another popular fractal-shaped antenna is the Koch snowflake. To construct a Koch snowflake, start with a filled equilateral triangle and construct a von Koch curve on each side of the triangle to obtain the geometry (iteratively), as shown in Figure 19, where the first three stages in the construction of a Koch snowflake are shown.

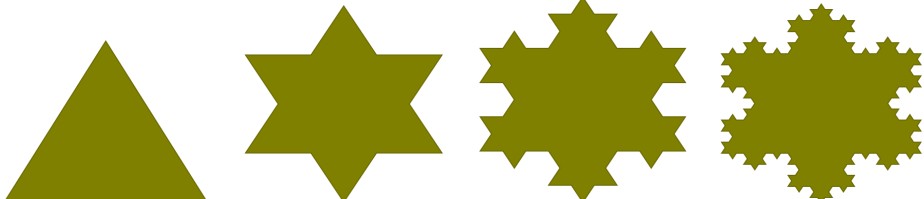

**Figure 19.** Four stages of the Snowflake fractal.

### 4.3.4. Minkowski Island Fractal Antenna

The construction of a Minkowski island fractal antenna is shown in Figure 20. Start with a filled square (called initiator). Then, replace each of the four sides of the initiator with the generator (shown at the bottom of Figure 20) and replace the four sides of the square with the generator and keep iterating. The result of this process is the Minkowski island fractal with intricate fundamental structure, which is nowhere differentiable.

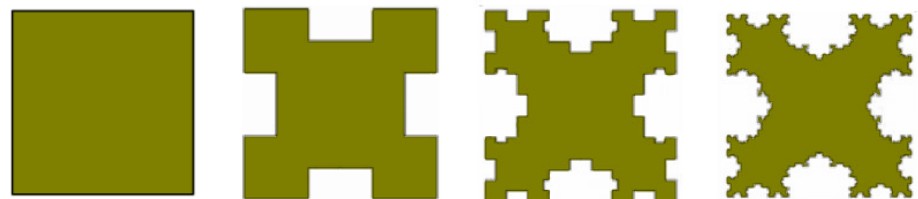

**Figure 20.** Four stages of the Minkowski fractal.

The Koch snowflake and Minkowski island fractal antennas have been extensively used to create new designs for miniaturized loops as well as microstrip patch antennas.

### 4.3.5. Hilbert Curve Antenna

The Hilbert fractal antenna is another type of wire antenna made from a space-filling curve and falls into the broad category of space-filling fractal antennas. The first four iterations in the construction of the Hilbert curve are shown in Figure 21. The Hilbert curve has properties such as self-avoidance (no intersection points), self-similarity, space filling and simplicity.

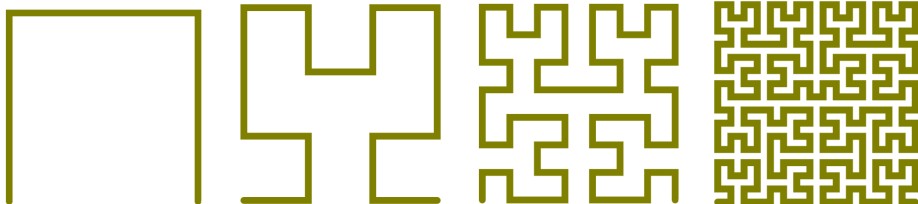

**Figure 21.** Four stages of the Hilbert fractal.

The space-filling properties of the Hilbert curve and related curves (e.g., Peano curves) make them suitable candidates for the design of fractal antennas.

### 4.3.6. Sierpinski Carpet or Fractal Pifa

An inverted-F antenna is another type of antenna first proposed by Ronold King at Harvard in 1958 for use in wireless communications. King's antenna was also a wire antenna and was designed for military use. It consists of a monopole antenna running parallel to a ground plane and grounded at one end.

Today, many cell phones comes with a Planar Inverted-F Antenna (in short PIFA), which are small, low profile, and sensitive to both horizontal and vertical polarized radio waves (see Baliarda et al. [19]), but the drawback is that PIFA are narrowband.

To overcome this difficulty, the fractal-shaped PIFA shown in Figure 22a has been designed, and the results are promising. A Fractal PIFA works similar to a traditional PIFA except that its design is a fractal based on a 2D Cantor array. A perfect fractal PIFA would be obtained by iterating the Cantor array an infinite number of times, but for practical design, two to three iterations are enough. A fractal PIFA mounted on a candy bar phone is also shown in Figure 22b, and a double PIFA is presented in Figure 22c.

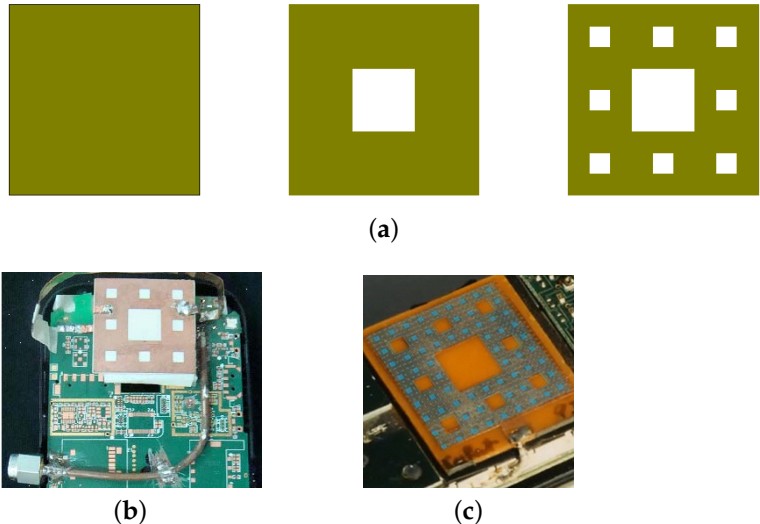

**Figure 22.** PIFA Antennas, (**a**) Three stages of Cantor fractal PIFA, (**b**) F-PIFA mounted on the candy bar phone, (**c**) Double-PIFA antenna.

### 4.3.7. Fractal Tree Antenna

Fractal trees antenna are used to fabricate miniaturized dipole antennas, and a number of new design of fractal tree antennas have evolved. An example of a ternary (three-branch) fractal tree is shown in Figure 23b, which looks like an analogue of the Sierpinski gasket of Figure 14. In fact, the ternary fractal tree shown in Figure 23a can be interpreted as a wire equivalent model of the Stage 4 Sierpinski gasket of Figure 14.

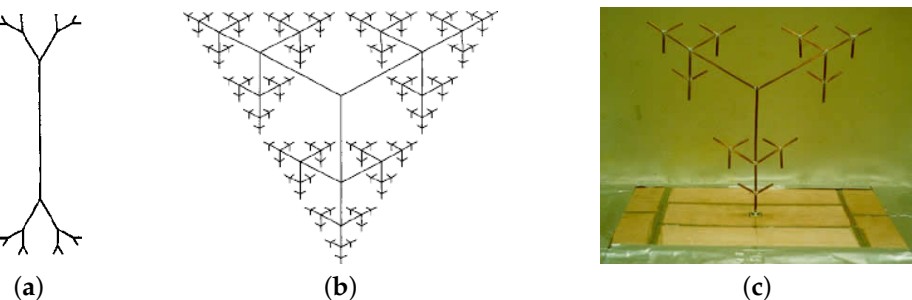

(a)  (b)  (c)

**Figure 23.** Fractal Tree Antennas: (**a**) Fractal tree, (**b**) A Stage 4 ternary fractal tree (Image source: Werner and Ganguly [15]), and (**c**) A prototype Tri-band fractal ternary tree monopole antenna used in miniaturized dipole antennas (Image source: http://cearl.ee.psu.edu/projects/project2-1-1.html, accessed on 22 June 2022).

We refer to the early papers by Werner [26] and Petko and Werner [27] for new designs and a variety of 2D and 3D multiband fractal tree antennas based on Koch curve and fractal trees, which are also reconfigurable (i.e., tunable) and exploit the self-similar branching structure of 3D fractal trees.

### 4.3.8. Other Innovative Fractal Antenna Designs

A multiband Cantor fractal monopole antenna covering GSM, DCS, PCS, UMTS, and WLAN applications was presented in [28].

A complementary stacked patch antenna based on Sierpinski fractal was introduced in [29], which enhanced antenna performances, retaining the basic characteristics of the Sierpinski antenna. A design procedure for custom made fractal antennas using artificial neural networks and the particle swarm optimization (PSO) was presented in [30]. A compact multiband E-shape fractal patch antenna was proposed in [31] multiband applications to achieve size reduction and increase the operating bands. This antenna operates on LTE/WWAN (GSM850/900/1800/1900/UMTS/LTE2300/2500) bands.

At present, many fractals are being used as antennas, and several patents are also registered on new discoveries. Some of the fractal antennas used in mobile phones are shown below in Figure 24. A microstrip patch antenna with edges in the shape of a Minkowski island fractal is shown in Figure 24a, which is used in iphones. The Sierpinski fractal carpet shown in Figure 24b was designed by the Spanish company FRACTUS as a built-in antenna for a GSM 900/1800 mobile handset.

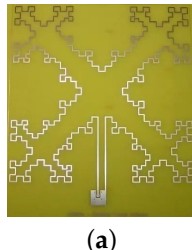 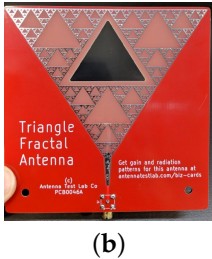

(a)  (b)

**Figure 24.** Some commercial antennas used in mobile phones and other applications, (**a**) Microstrip patch antenna, (**b**) Sierpinski triangle antenna.

Table 2 gives a summary of the literature for some standard fractal antennas and their modifications where antenna size(s), band utility, gain and applications are shown. It is clear from the table that the focus of designs is on multibandness with higher gain and effective bandwidth utility. Notice that reducing the dimensions of the designed antenna helps in miniaturization.

**Table 2.** Summary of the performance of some fractal antennas.

| Antenna Type (Ref. No.) | Dimension (mm$^2$) | No. of Bands | Bands (GHz) | Gain (dB) | Applications |
|---|---|---|---|---|---|
| Modified Sierpinski Gasket [32] | 27 × 29 | 1 | 3.16–9 | 9.00 | WLAN, WiMAX, public safety band, point to point high-speed applications for high data rates |
| Modified Sierpinski Gasket [33] | 30 × 34.64 | 2 | 12.2–13.4 21–30 | 21.20 8.00 | Broadband satellite receivers, mobile space research activities, active sensors, passive sensors |
| Modified Sierpinski Carpet [34] | 29.44 × 38.04 | 6 | 4.285 5.455 6.265 6.805 8.02 9.145 | – | Radio telecommunication in *C*-band, space communications in *X*-band and satellite communication |
| Modified Sierpinski Carpet [35] | 30 × 30 | 6 | 2.23 4.75 5.23 6.61 6.79 9.58 | 15.27 (max) | *S* (2–4 GHz) band, *C* (4–8 GHz) band Weather radar and satellite applications, etc. |
| Koch Snowflake [36] | 28.8 mm (diameter) | 1 | 3.34–4.52 2.2–3.4 1.45–4.1 | 3.30 (max) | Wideband applications |
| Koch Snowflake [37] | 60 mm (length of equal sides) 70 mm (base) | 5 | 11.44 13.178 15.482 19.902 23.529 | – | *X*-band, *Ku*-band and *K*-band |
| Minkowski Fractal [38] | 27.5 × 25 | 1 | 1.575 | 0.369 | Satellite Receiver |
| Hilbert Curve [39] | 49 × 52 | 4 | 0.876 1.225 1.850 2.400 | – | WSN Europe GPS-L1 GSM1800 Wi-Fi |
| Hilbert Curve [40] | 56 × 39.4 | 2 | 12.5–37.5 0.4–1.4 | 3.35 | HF/UHF dual band operation |
| Koch Curve Fractal Defected Ground Structure [41] | 1994.02 | 1 | 1.492–1.518 | 5.41 | *L*-band |
| Dual-Reverse-Arrow Fractal [42] | 46.4 mm (side length of triangle) | 1 | 2.4 | 2.5 | ISM Applications |
| Sierpinski Carpet and Minkowski Hybrid [43] | 40 × 40 | 2 | 3.5 5.8 | 4.50 | WiMAX LTE |
| Hetero Triangle Linked Hybrid Web Fractal [44] | 12 mm (diameter) | 1 | 1.945–20 | 7.17 (max) | 3G, LTE, ISM, Bluetooth, Wi-Fi, WLAN, WiMAX, Satellites (Ku-Band), etc. |

### 4.4. Fractal Metamaterials

Metamaterials are synthetic electromagentic materials having properties not found in standard conducting materials. These artificial composites inherit their properties from internal micro and nanostructures rather than the chemical composition as compared with natural materials.

Figure 25 shows the first manufactured fractal metamaterial invented by fractenna.com (which also holds a patent on this discovery). Fractal metamaterials can achieve wideband and multiband performance in the fields of cloaking, shielding, absorption and conveyance, whereas conventional metamaterial technology is limited to narrow passbands. This wideband/multiband performance is the key to employ fractal metamaterials in commercial and government applications. The field of fractal metamaterials is in the developmental stages, and their applications are still emerging.

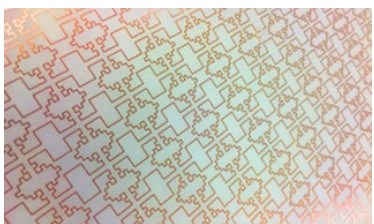

**Figure 25.** The first manufactured fractal metamaterial. (Image source: https://www.fractenna.com/, accessed on 22 June 2022).

### 4.5. Commercialization of Fractal Antennas

**(1) www.fractenna.com, accessed on 22 June 2022**

Dr. Cohen co-founded Fractal Antenna Systems, Inc. in the year 1995 to deliver the world's first fractal-based commercial antenna solutions (see Figure 26). Over the last 25 years, the company has deepened the theory of fractal antennas and deployed fractal antennas in a vast range of commercial and government applications. The company is also working on the capabilities and benefits of fractal metamaterials.

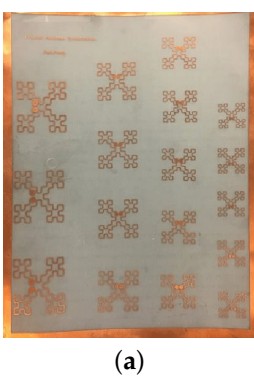 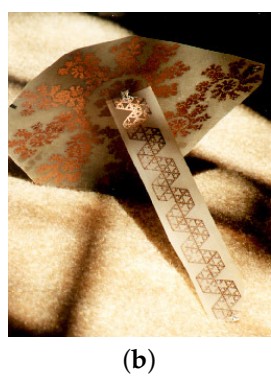

(**a**)          (**b**)

**Figure 26.** The first manufactured fractal antenna sheets (1995). Image source: https://www. fractenna.com/, accessed on 22 June 2022.

**(2) www.fractus.com, accessed on 22 June 2022**

Fractus is an early pioneer in the design and development of fractal antennas for smartphones, tablets, wireless and IOT devices. It was founded by fractal antenna pioneers Dr. Carles Puente and Ruben Bonet in 1999 and is leading the world market for its research, innovations and commercialization of multiband and miniature fractal antennas. The company holds the recognition of the world's first application for a patent on fractal and MultiFractal antennas.

**(3) Fractus Antennas S.L. (www.fractusantennas.com), accessed on 22 June 2022**

Founded in 2015, Fractus Antennas SL is actively involved in designing, manufacturing and commercializing miniature chip antennas for smartphones, short-range wireless and connected IoT devices. The company has received many patents for novel antenna designs. The recently developed Virtual Antenna™ Technology (2019) by Fractus SL is so unique that each antenna can be used for any application such as GSM (2G, 3G, 4G, 5G), GPRS, GPS, Bluetooth, WI-FI, RFID, NB-IOT, NBLTE and many more.

### 4.6. Summary

Fractal antennas are a replacement for traditional wideband/multiband antennas that are smaller and lighter, require less circuitry, have fewer radiative elements to resonate at multiple frequencies and provide higher gains. Antennas with fractal shapes have many possible applications ranging from dual-mode phones to location services such as GPS, satellites, etc. Fractal-shaped antennas can lower the radar cross-section (RCS), which can be exploited in military applications where the RCS is an extremely important design parameter.

In the future, fractal antennas will play a much bigger role in the developing technologies for wireless communications which require compact, wideband and multiband antennas. Examples include wireless devices such as cell phones, tablets, wearable devices, smart homes, smart cities, airplanes, and IoT devices. The design of a high-performance wideband antenna is critical to IoT and wireless connectivity, and the fractal antenna engineering is enabling the changes that are required.

## 5. Fractals in Image Compression

The need for mass information storage and retrieval is growing rapidly with the advancement of the data and information age. On a computer, images are stored as a collection of bits representing pixels. Storing a single image or a collection of images on a computer may require large memory. This problem can be addressed using various image compression techniques. Storing images in less memory leads to a direct reduction in cost. This is where image compression plays an important role. Another useful feature of image compression is rapid data transfer, since less data need less time to transfer.

The Discrete Cosine Transform Algorithm is one of the most popular image compression methods, which is used in JPEG (still images), MPEG (motion video images), H.26x digital audio (such as Dolby Digital, MP3, AAC), and television (SDTV, HDTV) compression algorithms.

Fractal image compression is a fractal-based compression technique that makes use of the self-similarity present in an image for fractal coding. It is simple to implement, easy to execute and yields high compression ratios and quick decompression. Fractal image compression (FIC) was introduced by M. Barnsley, who started a company based on FIC technology. However, it was Arnaud Jacquin (a doctoral student of Barnsley) who published a fractal image compression algorithm for the first time.

### 5.1. History of Fractal Image Compression

After Mandelbrot's pioneering work [2], John Hutchinson introduced the iterated function theory in 1981 as an answer to the search of an underlying mathematical framework for fractal geometry. Later, M. Barnsley, another leading researcher in developing a mathematical framework for fractal geometry, wrote the famous book *Fractals Everywhere*. In this book, Barnsley described Iterated Functions Systems (IFS) and a very useful result known as the *Collage Theorem*, which became a fundamental result for fractal image com-

pression. For example, the Pythagorean tree in the Figure 27 can be generated using the two-dimensional IFS

$$f_1 \begin{pmatrix} x_1 \\ x_2 \end{pmatrix} = \begin{bmatrix} 1/2 & -1/2 \\ 1/2 & 1/2 \end{bmatrix} \begin{pmatrix} x_1 \\ x_2 \end{pmatrix} + \begin{pmatrix} 0 \\ 1 \end{pmatrix}, \tag{4}$$

$$f_2 \begin{pmatrix} x_1 \\ x_2 \end{pmatrix} = \begin{bmatrix} 1/2 & 1/2 \\ -1/2 & 1/2 \end{bmatrix} \begin{pmatrix} x_1 \\ x_2 \end{pmatrix} + \begin{pmatrix} 1/2 \\ 3/2 \end{pmatrix}, \tag{5}$$

$$f_3 \begin{pmatrix} x_1 \\ x_2 \end{pmatrix} = \begin{bmatrix} 1 & 0 \\ 0 & 1 \end{bmatrix} \begin{pmatrix} x_1 \\ x_2 \end{pmatrix}. \tag{6}$$

Iterated function systems produce attractors (fractals), which are fixed points of a contraction mapping defined using the IFS, and the collage theorem does the reverse; i.e., for a given initial image, find an IFS whose attractor is as close as possible to the given image.

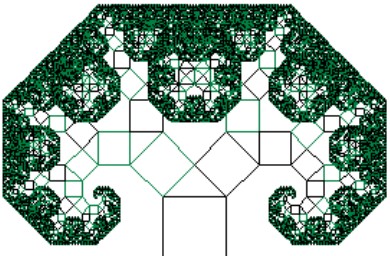

**Figure 27.** Pythagorean tree constructed using IFS in Equations (4)–(6).

Michael Barnsley suggested that storing images (for instance, the fractal tree shown in Figure 27) as a set of transformations given in Equations (4)–(6) may lead to image compression. IFS is a set of transformations from which the image of an attractor can be obtained. Barnsley did it in reverse by generating an IFS of the image which maps onto itself by making use of the collage theorem [6]. This leads to the compression of images. Barnsley observed many affine redundancies in real-life images and noticed that memory can be saved if we store suitable IFS. He was granted a patent and co-founded Iterated Systems Incorporation along with Alan Sloan. Barnsley published his results in the January 1988 issue of the BYTE magazine. This article exhibit several images compressed in excess of 10,000:1. The images were named as "Black Forest", "Monterey Coast" and "Bolivian Girl", but they were all manually constructed. Barnsley's patent is referred to as the "*graduate student algorithm.*"

In March 1988, Arnaud Jacquin found a modified scheme for representing images called *Partitioned Iterated Function Systems* (PIFS) that made the graduate student algorithm obsolete. In 1991, Barnsley gave another algorithm that can automatically convert an image into a PIFS, compressing the image in the process, and he received another patent for this. All contemporary fractal image compression algorithms are based on Jacquin's algorithm, and attempts to improve it have continued to date.

### 5.2. Mathematics of Images

Mathematically, an image is expressed as a function $z = f(x, y)$, where $z$ is the grayscale. We define the distance between two images $f(x, y)$ and $g(x, y)$ by the metric

$$d_{\max}(f, g) = \max_{(x,y) \in P} |f(x,y) - g(x,y)|, \tag{7}$$

where $f$ and $g$ are values of the level of gray pixel (for grayscale image), $P$ is the space of images, and $x, y$ are the coordinates of any pixel. It is clear from (7) that the $d_{\max}$ metric searches for the point $(x, y)$ at which the two images $f$ and $g$ differ the most and assigns

this as the distance between $f$ and $g$. Another useful metric used in image compression is the root mean square (rms) metric (more useful for practical calculations) defined by

$$d_{\text{rms}}(f,g) = \sqrt{\int_P (f(x,y) - g(x,y))^2 dxdy}. \tag{8}$$

Grayscale images are representations of subsets of the plane. An image is represented as a collection of pixels, and an image containing $m \cdot n$ pixels can be regarded as a vector in $r = m \times n$ dimensional space. Typically, the space is $\mathbb{R}^2$, and the usual norm on $\mathbb{R}^2$ is the 2-norm (also called the Euclidean norm or the $L_2$ norm), which is defined by

$$||x||_2 = \sqrt{|x_1|^2 + |x_2|^2}, \tag{9}$$

which induces the rms metric,

$$d_{\text{rms}}(x,y) = ||x - y||_2.$$

Thus, if $x = (x_1, \ldots, x_r)$ and $y = (y_1, \cdots, y_r)$ are images, then the $L_2$ norm or rms distance (gap) between them is given by

$$d_{\text{rms}}(x,y) = ||x - y||_2 = \sqrt{\sum_{i=1}^{r} (x_i - y_i)^2}. \tag{10}$$

Fidelity (a measure of the correctness of the reconstructed image) of an image is computed using the root mean square error ($e_{\text{rms}}$), the signal to noise ratio (SNR) and the peak signal to noise ratio (PSNR) of the image. Let $I(x,y)$ and $A(x,y)$, respectively, denote the gray levels on the original and the reconstructed image (attractor), respectively; then,

$$e_{\text{rms}} = \sqrt{\sum_{x=1}^{m} \sum_{y=1}^{n} (e(x,y))^2}, \ \ e(x,y) = (I(x,y) - A(x,y)), \tag{11}$$

$$SNR = \frac{\sum_{x=1}^{m} \sum_{y=1}^{n} (A(x,y))^2}{\sum_{x=1}^{m} \sum_{y=1}^{n} (e(x,y))^2}, \quad PSNR_{\text{rms}} = 20 \log_{10}\left(\frac{2^p - 1}{e_{\text{rms}}}\right), \tag{12}$$

where $p$ is the number of bits per pixel used for definition of the gray level.

### 5.3. Self-Similarity in Target Images

In general, a typical image does not show exact self-similarity, which is seen in mathematical fractals. However, it still contains a type of self-similarity in the sense that the entire image may not be self-similar, but parts of the image are self-similar with properly transformed parts of itself. For example, Figure 28 shows some parts of the Lena image that are self-similar at different scales.

A portion of the reflection of the hat in the mirror is similar to a smaller part of her hat, and a part of her shoulder overlaps a smaller region that is almost identical. Studies [2,45,46] suggest that most of the natural images contain this kind of self-similarity. The search for the resemblance (self-similarity) is the base of fractal compression algorithms.

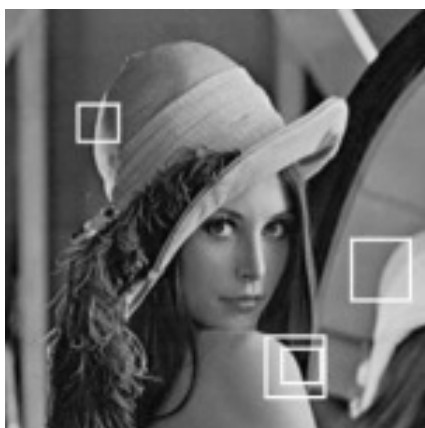

**Figure 28.** Self-similarity in the Lena image.

### 5.4. Classical Approach

Imagine a Multiple Reduction Copying Machine (MRCM) shown in Figure 29. A MRCM (with multiple lens arrangements) is just like a regular copying machine except that it will scale the original image (to be copied) by half and print it three times on the copy.

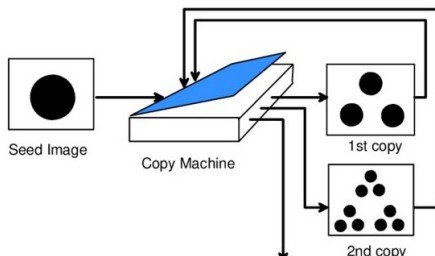

**Figure 29.** A Multiple Reduction Copying Machine (MRCM) with sample outputs. Reprinted with permission from [47]. Copyright 1997 Springer.

Figure 30 shows a few iterations of feeding an input (a Mandelbrot image) to the machine, and on repeated back feeding the output as input, the final image (attractor) is the Sierpinksi triangle.

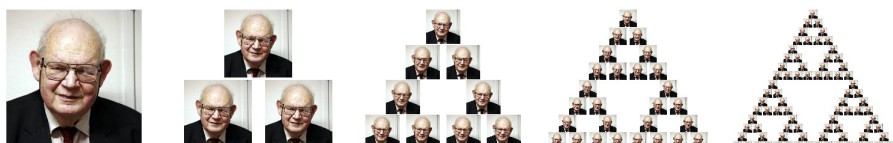

**Figure 30.** The first 4 copies of an input image generated by the MRCM of Figure 29.

Clearly, any initial image will shrink to a point on repeated iterations due to size reduction in every iteration on the photocopying machine. Therefore, the shape of the final image (attractor) is determined by the position and the orientation of the image and not by its initial size.

In fractal image compression, to encode an image $f$, we need to find the transformations $w_1, w_2, \ldots, w_n$ such that $f$ is the attractor of the map $W = \bigcup_{i=1}^{n} w_i$. Thus, we partition the image into pieces, find the transformations $w_i$, and acquire the original image $f$ again by applying the transformations $w_i$.

The final output from the photocopying machine is determined by the way in which an input image is transformed by the transformations $w_i$ when running the machine in a feedback loop. Theretofore, the transformations must be contractive; that is, each of these

transformations must bring any two points of the input image closer in the output. In practice, it is sufficient to choose affine transformations of the form

$$w_i \begin{pmatrix} x_1 \\ x_2 \end{pmatrix} = \begin{pmatrix} a_i & b_i \\ c_i & d_i \end{pmatrix} \begin{pmatrix} x_1 \\ x_2 \end{pmatrix} + \begin{pmatrix} e_i \\ f_i \end{pmatrix}, \quad i = 1, 2, \cdots, n. \tag{13}$$

Each transformation can rotate, scale (shrink) and translate an input image. Each $w_i$ is a contraction mapping as long as the determinant of the transformation is strictly less than one, and the IFS will converge to the attractor $A$ starting with any image $A_0$. Indeed, we have

$$A = \lim_{n \to \infty} W^n(A_0), \quad \text{with} \quad W(A) = \bigcup_{i=1}^{N} w_i(A). \tag{14}$$

In Figure 30, the final image obtained on repeated application of the transformation $W$ possesses geometric self-similarity, and that is why IFSs are always expected to generate fractal images.

### 5.5. Contemporary Approach

The basic idea of partitioned iterated function system (PIFS) is as follows: if finding self-similarity between an image as a whole and its parts is impractical, then finding self-similarity between larger and smaller parts of the image is more reasonable. Using Jacquin's approach, this can be done by partitioning the original image at different scales into larger parts called *domain blocks* and small parts called *range blocks*. The idea of the PIFS is illustrated in Figure 31.

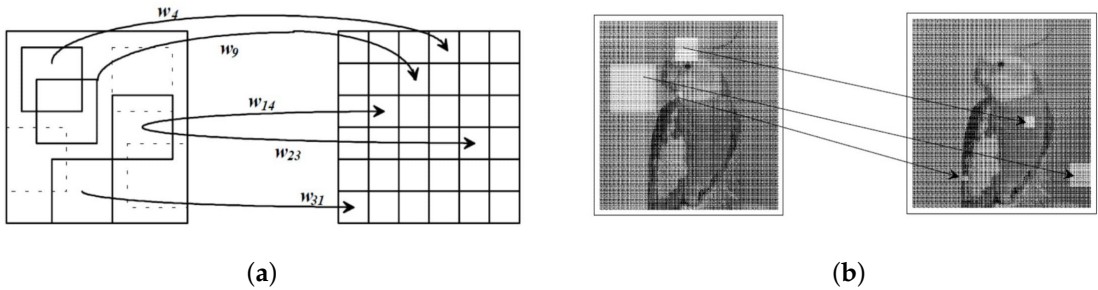

(a)　　　　　　　　　　　　　　　　　　　　(b)

**Figure 31.** Self-similarity in Partitioned Iterated Function System, (**a**) Domain (left) and Range (right) blocks, (**b**) Domain–range pair self-similarity at three scales. Reprinted with permission from [47]. Copyright 1997 Springer.

where some mappings from domain blocks to range blocks are shown. The range blocks are disjoint and partition the image uniformly. The domain blocks may overlap and need not contain every pixel of the original image. The goal of the compression process is to find a closely matching pre-image (i.e., a domain block for every range block). The size of the *domain pool* (determined by the number of domain blocks) is important for the encoding purpose. In general, a larger domain pool implies better fidelity of the mappings between the domain blocks and the range blocks. However, this also leads to more comparisons, which slow down the encoding. A scheme for classifying the domain and range blocks can be found in [46,48].

### 5.6. Partitioned Iterated Function System

Jacquin extended the definition of an IFS to Partitioned Iterated Function Systems (PIFS) [48] in an attempt to ease the IFS computations. Theoretically, each image has a unique fixed point, but it is not feasible to find a unique fixed point for a whole image in practice. Thus, as an alternative, the image should be partitioned into several parts, and the fixed points for each part should be obtained through different transformations. We

will use only affine transformations to illustrate a PIFS for simplicity, although the PIFS is independent of the type of transformations used. There are two spatial dimensions $x$ and $y$, and the gray level adds a third dimension to the IFS so that the modified affine transformation $w_i$ for PIFS becomes

$$w_i \begin{pmatrix} x \\ y \\ z \end{pmatrix} = \begin{pmatrix} a_i & b_i & 0 \\ c_i & d_i & 0 \\ 0 & 0 & s_i \end{pmatrix} \begin{pmatrix} x \\ y \\ z \end{pmatrix} + \begin{pmatrix} e_i \\ f_i \\ o_i \end{pmatrix}. \tag{15}$$

To achieve convergence, the intensity value of a pixel must be scaled and offset, i.e.,

$$z' = s_i z + o_i. \tag{16}$$

Here, $x$ and $y$ are the spatial locations of a pixel, while $z$ is the gray-level intensity of the pixel at location $(x, y)$. Coefficients $a_i, b_i, c_i, d_i, e_i$ and $f_i$ control skewing, stretching, rotation, scaling, and translation, while the coefficients $s_i$ and $o_i$ determine the contrast and brightness of the transformation, respectively, which allow the affine transformation to map grayscale domain blocks to grayscale range blocks accurately (see Figure 31b for three examples).

To speed up the compression and bring it under control, Jacquin constrained Equation (15) so that the domain blocks are always squares and equal to two times the size of range blocks. For instance, if the range blocks are (say) $8 \times 8$ pixels in size, then the domain blocks are chosen to be of the size of $16 \times 16$ pixels, which reduces the number of domain blocks to a large extent, and the search time is reduced during compression.

Thus, the image can be represented as a union of maps $w_1, w_2, \ldots, w_N$, such that $w_i : D_i \to \hat{R}_i$. That is, the application of $w_i$ to a region of the image $D_i$ produces $\hat{R}_i$, which is a result that approximates another region of the image, $R_i$. Minimizing the error between $\hat{R}_i$ and $R_i$ will minimize the error between the original image and the approximation. In practice, the RMS metric is used to find the "best" transform to map $D_i$ to $R_i$.

*5.7. The Encoding*

To encode a given image $f$, our aim is to find transformations $w_1, w_2, \ldots, w_n$ such that $f$ is the fixed point of the map $W$. In other words, we decompose $f$ into parts, apply the transformations $w_i$, and recover the original image $f$.

Fractal coding can produce a high compression ratio, which makes it one of the main advantages in compressing images. In Jacquin's algorithm, the aim is to minimize the Hausdorff distance (i.e., greatest pixel-to-pixel difference) between a candidate domain block and a specific range block.

The optimal scaling parameters can be computed algebraically if the root mean square error measure is used. To see this, assume that the domain block $D_{xy}$ has been reduced to the size of the range block $R_{xy}$. Then, the mean square error between the blocks is

$$e_{\text{rms}} = \frac{1}{n^2} \sum_{x=1}^{n} \sum_{y=1}^{n} (s_i D_{xy} - R_{xy})^2. \tag{17}$$

setting the derivative equal to zero

$$\frac{\partial e_{\text{rms}}}{\partial s_i} = \frac{2}{n^2} \sum_{x=1}^{n} \sum_{y=1}^{n} (s_i D_{xy} - R_{xy}) D_{xy} = 0, \tag{18}$$

we obtain

$$s_i = \frac{\sum\limits_{x=1}^{n} \sum\limits_{y=1}^{n} R_{xy} D_{xy}}{\sum\limits_{x=1}^{n} \sum\limits_{y=1}^{n} (D_{xy})^2}. \tag{19}$$

Figure 32 displays the flowchart of the encoding process.

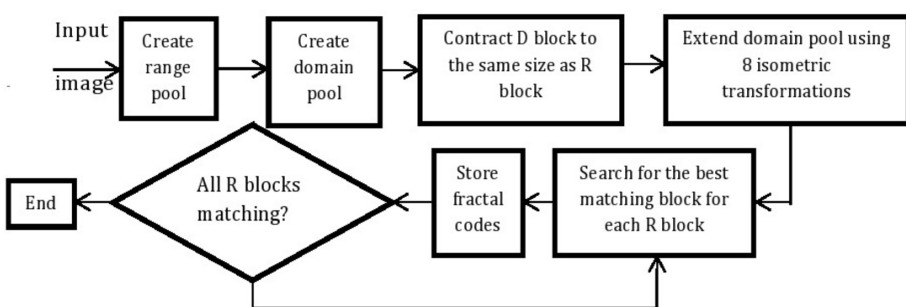

**Figure 32.** Encoding process.

Consider, for example, an image of size $128 \times 128$ pixels such that each pixel is of 256 gray levels. The image is partitioned into $8 \times 8$ blocks of non-overlapping range blocks and $16 \times 16$ overlapping domains blocks. For each range block $R_i$, a search is done through the entire set of domain blocks $D$ to find the domain block which matches best with $R_i$. The position of the range, the best matching domain block, and transformation $w_i$, which minimizes the distance between domain and range blocks, are stored. This process is repeated until we have found the best matching block for the domain–range pair. This method of partition is a fixed range size partition method.

Table 3 shows the results of this process on the compression and reconstruction of 13 images using the classical approach [45,49].

**Table 3.** Performance of the Barnsley's algorithm [45] on various images.

| Image Name | $\dfrac{\text{Time}}{\text{Time Average}}$ | $e_{rms}$ | $SNR_{rms}$ | $PSNR$ (dB) |
|:---:|:---:|:---:|:---:|:---:|
| Lena | 1.000200294 | 7.61672 | 13.3478 | 30.4954 |
| Peppers | 1.000266348 | 7.5512 | 11.9571 | 30.5705 |
| Mandril | 1.000238648 | 13.168 | 1 8.919 | 25.7403 |
| LAX | 1.00021521 | 17.4734 | 4.9517 | 23.2832 |
| Cameraman | 1.000095885 | 14.0104 | 8.1885 | 25.2018 |
| Columbia | 1.000159809 | 16.3936 | 5.3475 | 23.8373 |
| Goldhill | 1.000138501 | 6.79771 | 13.4355 | 31.4836 |
| Couple | 1.000006392 | 13.6817 | 8.3402 | 25.408 |
| Plane | 0.999230786 | 13.2835 | 7.5457 | 25.6646 |
| Women | 1.000091624 | 11.0847 | 10.2303 | 27.2363 |
| Milk | 1.000093755 | 9.6735 | 8.4824 | 28.4191 |
| Man | 1.000025569 | 11.7124 | 9.3384 | 26.7579 |
| Lake | 0.999232916 | 15.8357 | 2.9538 | 24.138 |
| **Average** | 1.000000 | 12.1756 | 8.69527 | 26.7874 |

All images are of size $128 \times 128$ pixels (=16384 pixels) and 256 gray level. The range blocks are $4 \times 4$ pixels, and the domain blocks are $8 \times 8$ pixels. Therefore, the number of blocks to be encoded is $\left(\frac{128}{4}\right)^2 = 1024$. For the purpose of comparing image quality on the reconstruction of these 13 pictures, we refer to [49].

*5.8. Decompression Process*

The decoding process involves repeatedly applying the transform until it converges to an image, which closely approximates the original image. The decompression starts by setting the image buffer to a uniform mid-gray value, which is used as the seed image, and the pixels of each range block in the transform list are evaluated during the iteration. The result of the first iteration is used as input for the second stage of iteration. Usually, the original image is recognizable in just two iterations, and typically, the decompression process will converge in four or five iterations (when 8-bit precision is used per pixel). The

decompression process for two encoded grayscale images of a 'Bird' and a 'Cameraman' is shown in Figure 33.

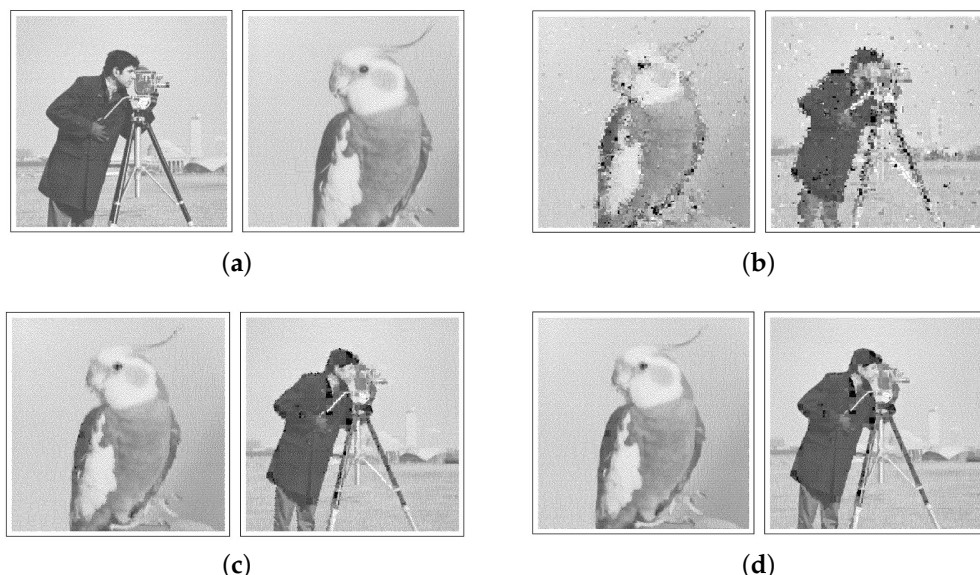

(a)  (b)

(c)  (d)

**Figure 33.** Decompression process for Bird and Cameraman (Reprinted with permission from [47]. Copyright 1997 Springer), (**a**) Seed for Bird (left) and seed for Cameraman (right), (**b**) 2 iterations of Bird IFS (left), 2 iterations of Cameraman IFS (right), (**c**) 4 iterations of Bird IFS (left), 4 iterations of Cameraman IFS (right), (**d**) 6 iterations of Bird IFS (left), 6 iterations of Cameraman IFS (right).

The choice of seed image has no impact on the outcome, since the IFS in Equation (14) describes the same attractor regardless of the starting image. This fact is well observed in Figure 33, where the Cameraman image is used as the seed image for Bird, and the Bird image is used as the seed image for Cameraman (see Figure 33a). One can notice the defects in Figure 33b which result from choosing a 'wrong' initial image that ultimately disappear with increasing iterations. The choice of seed can affect the decompression time, though, and it can be verified by starting with an all-black seed or an all-white seed image. However, for practical purposes, a mid-gray or a low-resolution version of the original image is preferred as the seed. See Figure 34 for a comparison of convergence using various seed images.

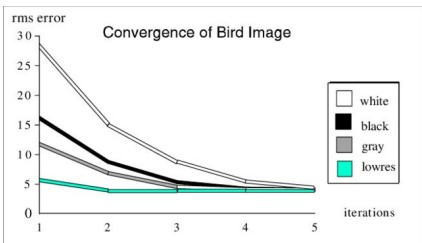

**Figure 34.** Convergence speed for various seed images. Reprinted with permission from [47]. Copyright 1997 Springer.

### 5.9. Partitioning Schemes

Partitioning of an input image is an important aspect of fractal image compression. Image partitioning refers to dividing the image into sections that are more appropriate for the application to work on.

In the classical approach of Jacquin [48], the image is partitioned into a fixed size square range blocks and domain blocks in which the size of domain blocks is twice the size of the range blocks. Several other flexible partitioning methods have evolved over the years, which allow for a higher compression ratio and shorter encoding times. Fisher [46]

introduced the quadtree, HV Partitioning and Triangular partitioning schemes shown in Figure 35. We also refer to the review paper by Wohlberg and Jager [50] for the details on various partitioning schemes studied in the literature. Among all partitioning schemes, the quadtree partitioning is the most widely used technique.

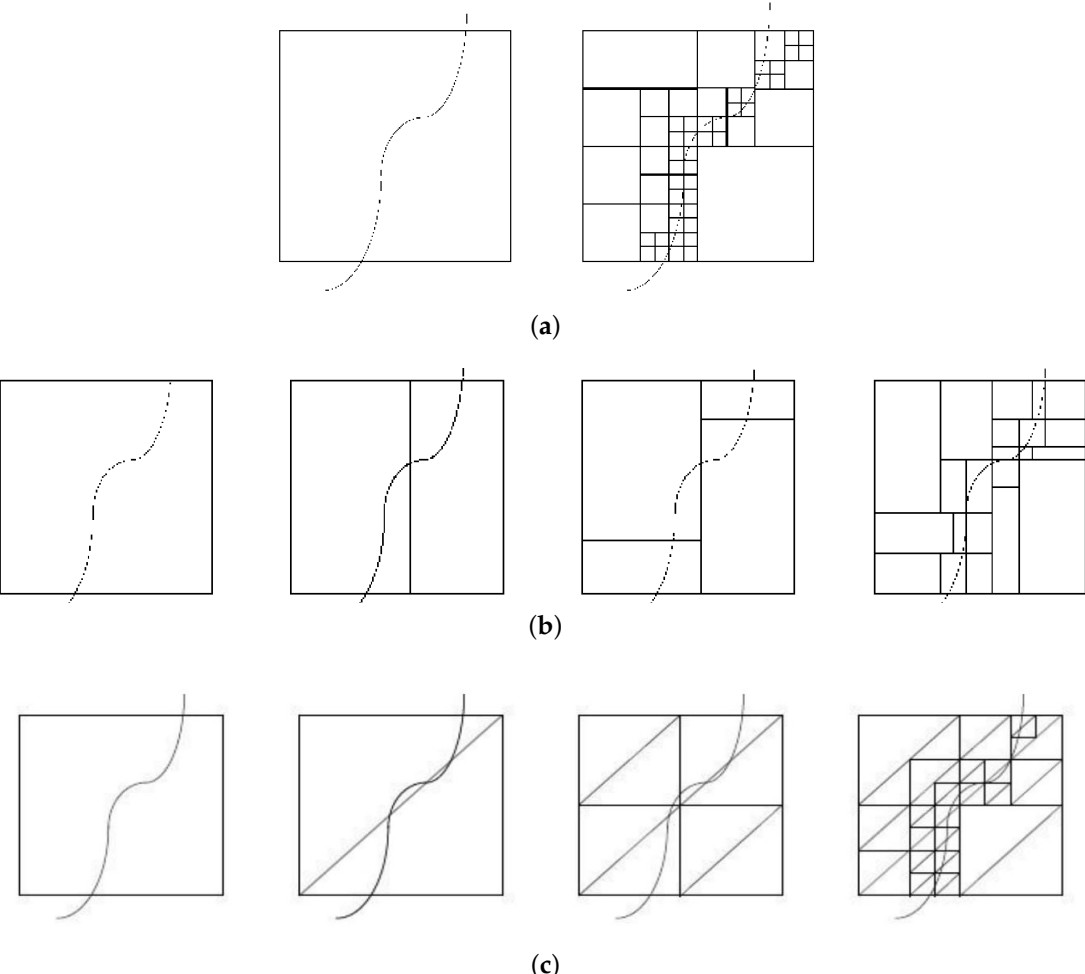

**Figure 35.** Some popular partitioning schemes, (**a**) Quadtree partitioning, (**b**) HV partitioning, (**c**) Triangular partitioning.

*5.10. Summary of Fractal Image Compression*

Fractal image compression is a promising, block-based, lossy and asymmetrical compression method. The images generated by fractal coding are *resolution/scale independent*, i.e., the image can be decoded at any resolution. Magnifying an image reveals additional detail, and after every iteration, details on the decoded image are sharper than before. This feature of fractal image compression is unique. Figure 36 shows magnification of the original image of Lena's eye (on the left). On the right is the same part of the fractal image rendered at the same scale. Sometimes, magnified fractal encoded images often look better than magnified original images due to reasonable interpolation.

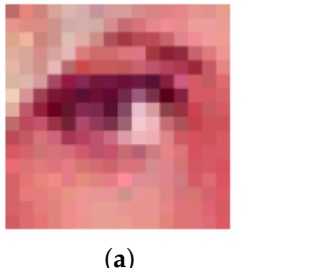 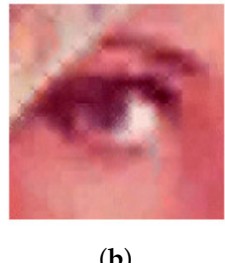

(**a**)                 (**b**)

**Figure 36.** Resolution independence: (**a**) Original image enlarged 4 times, (**b**) Decoded image enlarged 4 times

Another main advantage of FIC is that it is easy to automate. Decompression is quick, and fractal compression can achieve high compression ratios while maintaining image quality, and at higher compression, it is relatively superior to JPEG and wavelet compressions. Fractal compression is also useful in multimedia applications. Fractal compression methods are probably best suitable for archival applications, such as digital encyclopedias, where encoding is done only once. The greatest challenge for the coding community is how to precisely measure and quantify signal-to-noise ratio, root mean square error, etc.

Fractal image compression is still under development. Many research groups worldwide are developing new algorithms to shorten the encoding time. We refer the reader to [45,46,48,49,51] for more detailed literature on the theoretical concepts, existing methods, algorithms and experimental results on fractal image compression.

## 6. Fractals in Fracture Mechanics

Fracture mechanics is the study of the propagation of cracks in materials, and it is an important tool to improve the performance of mechanical components. The phenomenon of fracture is to divide an object or material into two or different pieces on applying physical stress (see Figure 37 for different types of fracture modes). Thus, there exists a crack on the surface irregularly which penetrates into the body, too. All these physical appearances such as crack length, area, etc. cannot be described easily using Euclidean geometry. The fractal geometry equipped with self-similarity (or self-affinity), scale invariance and fractal dimension offers great help to analyze irregular or fractional shapes of fracture mechanics.

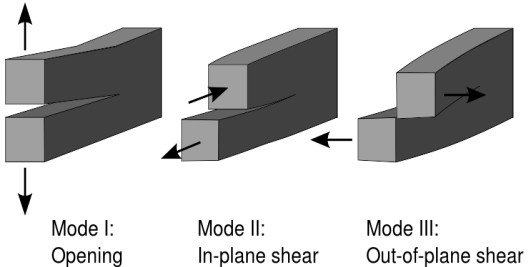

Mode I:
Opening

Mode II:
In-plane shear

Mode III:
Out-of-plane shear

**Figure 37.** The three fracture modes. (Image source: https://en.wikipedia.org/wiki/Fracture_mechanics, accessed on 22 June 2022).

Mandelbrot was the first to interrelate the crack propagation and other fracture properties of materials with the fractal geometry [52]. He introduced a method called *slit island analysis* on the fracture surface to find fracture dimensions, which is shown to be a measure of toughness in metals. Mandelbrot characterized the structure of a surface by the fractal dimension, *D*, as a scaling factor. As *D* increases from 0 to 1, the irregularities of the surface become more significant, and shape becomes predominantly less meaningful. He experimented through fractured steel specimen plated with electroless nickel and proposed the "slit island analysis" method to calculate the fractal dimension.

The quantitative analysis of fracture surfaces in brittle alumina and glass ceramic materials using fractal geometry was considered by Mecholsky et al. [53] by calculating the fractal dimension of crack surfaces using slit island analysis (SIA) and fracture profile analysis (FPA) methods. They proved that the fractal dimension increases with increase in fracture toughness, in general.

Fractal geometries are often characterized by a scaling (power) law:

$$Nr^D = 1. \tag{20}$$

where $N$ is the number of segments, $r$ is the similarity ratio (or reduction factor), and $D$ is the fractal dimension.

Equation (20) describes how many new features will appear by a magnification factor $r$ for a given fractal dimnesion. For example, if $r = \frac{1}{4}$ and $D = 1.5$, then the number of features will be $N = 8$. The number of features would increase to $N \approx 11$ at the same scale with $D = 1.75$. Thus, the higher fractal dimension leads to more features or structures.

The toughness of a fracture surface is measured in terms of difficulty in the crack growth, and researchers have attempted to relate the fracture toughness and surface energy with the fractal dimension. In this connection, Mecholsky et al. [54] discovered the following formula relating fractal dimension with the fracture toughness

$$K_{IC} = E(a_0 D^*)^{\frac{1}{2}}. \tag{21}$$

Here, $E$ is the modulus of elasticity of the material, $a_0$ is the lattice parameter, $D^* = D - d$ with $d$ as the Euclidean dimension in the projection of fracture. Mu and Lung [55] proposed an alternate equation which is a power law relation connecting the fractal dimension with surface energy.

Zhang [56] studied the fracture of rocks under the effect of high temperature considering the fractal dimension as a crucial factor. Fractal dimension and the rockburst tendency index can predict the failure of the rocks, and variations in rockburst tendency laws were been obtained. The relation between fractal dimension and rockburst tendency can be explained by a quadratic expression

$$K_{\text{eff}} = A(d_f - \bar{d}_f)^2 + B.$$

Here, $K_{eff}$ is the effective burst energy index, $A$ and $B$ are rock material constants, $d_f$ is the fractal dimension of the fracture surface and $\bar{d}_f$ is the fractal dimension threshold, and there is a directly proportional correlativity between rockburst index and the fractal dimension when $d_f \le \bar{d}_f$ and inverse proportionality correlation when $d_f > \bar{d}_f$. This is how mechanical properties such as energy dissipation energy release rates related with the fractal dimension of the fractured surface during the rock failure mechanism, and that will reflect in the degree of rockburst tendency.

After the pioneering work of Mandelbrot et al. [52], fractal geometry has been applied to the fractality of cracked surfaces, fracture mechanics and material science problems by several authors, and we refer to the papers [54,56–58] for further details, analysis and determination of the fractal dimension of microcrack structures and fracture surfaces.

## 7. Other Fractal Applications and Innovations

**Fractals in ophthalmology:** The human retina shown in Figure 38 exhibits fractal structure properties in its vascular network, so fractal geometry is the right tool for modeling such a complex structure [59]. The damage of the blood vessels of the retina in diabetic people is known as *diabetic retinopathy*.

The examination of fundus of the eye is a classical old technique for screening diabetic retinopathy and takes more time. In recent times, the technique of taking digital photographs of the fundus is used, which are transmitted to a central database for testing.

Fractal analysis is the best method in processing this data with more accurate results as compared to other methods where the fractal dimension is the prominent tool for analysis.

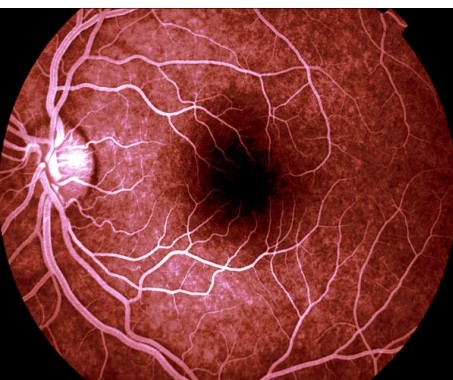

**Figure 38.** Human retina. Image by: Paul van der Meer. (Image source: https://fractalfoundation. org/OFC/OFC-1-3.html, accessed on 22 June 2022).

Fractals are also important in other life science studies and biological fields. They are now used to predict or analyze the growth patterns of bacteria, the pattern of nerve dendrites, pathology, study of cancer, wildlife and landscape ecology, etc. The expository article by G.A. Losa [60] is a rich source of information on the extension of fractal geometry for the life sciences to understand complex functional properties, morphological, and structural features characterizing cells and tissues. The reader may also refer to [61] and references therein for further study. In most of these studies, fractal dimension is a key tool for analysis.

**Fractal Capacitors:** Wearable and implantable electronic devices are common nowadays and are expected to dominate the future soon. However, these devices suffer the problem of inadequate power supply limited by the size of these gadgets. Microsupercapacitors (MSCs) are emerging miniaturized high-power microelectrochemical energy-storage devices that can circumvent this difficulty, as they are capable of delivering high power density, fast charge and discharge, and a superior lifetime (millions of cycles). In a recent study, Hota et al. [62] fabricated integrated MSCs using three different fractal designs—namely, Hilbert, Peano, and Moore (they used anhydrous $RuO_2$ thin-film electrodes as prototypes)—and proved that fractal-shaped electrode designs is a viable solution to improve the performance of MSCs. It is shown that among the three proposed designs, the Moore design shows the best performance. Many more MSCs may be fabricated by exploiting the self-similarity and scale invariance of fractals.

**Fractal Batteries:** Fractal structures have proven to be advantageous in electrochemical energy conversion systems, since fractals maximize the electrochemically active surface area while minimizing the energy loss in the network. Fractals can be used in the "fractalization" of battery electrodes to increase power density and reduce dendrite formation. The fractalization technique can be applied to any electrode material (e.g., C, Si, MgX, etc.). In this connection, we refer to [63], wherein the theoretical analysis of fractal type electrodes for lithium-ion batteries is presented along with simulation results. More recently, Thekkekara and Gu [64] proposed bio-inspired fractal electrode designs for solar energy storage using space-filling properties of fractal curves from the Peano family.

**Fractal Electromagnets:** The techniques of fractal geometries can be used to fabricate fractal electromagnets to increase the magnetic flux for a given size, or, alternatively, shrink the size for a given flux. This size reduction permits embedding electromagnets and solenoids in places where it was almost impossible until now.

**Fractal PCBs:** Fractals are being applied on printed circuit boards (PCBs) to reduce corrosion possibilities by fabricating fractal-shaped PCBs. Fractal PCBs can be applied to any trace or joints of contact with a high-voltage differential to reduce the risk of corrosion.

Less corrosion delivers high reliability in electrical components, resulting in reduced overall cost.

**Fractal in Cooling Devices:** Fractal-shaped smart cooling devices such as cooling chips, PC coolers, fractal microchannel heat sink, etc. are now becoming popular, which are based on fractal geometry. A cooling circuit for a computer chip printed in the form of a fractal branching pattern is shown in Figure 39a. The liquid nitrogen passes across the surface through this device to keep the chip cool.

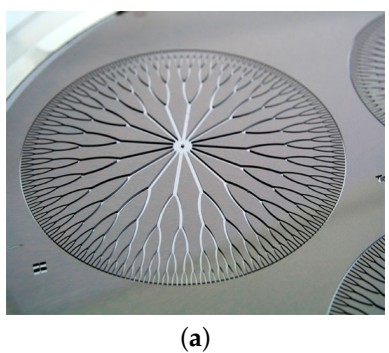
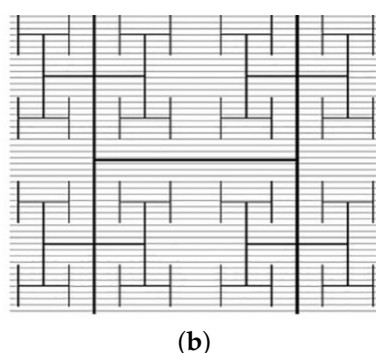

(**a**)                    (**b**)

**Figure 39.** (**a**) Computer chip cooling circuit, (**b**) A fractal solar panel [65].

**Fractal Solar Panels:** A group of researchers from the University of Oregon [65] have recently proposed a new electrode design based on the H-tree fractal tree structure (see Figure 39b) for fractal-patterned rooftop solar panels that combines the aesthetic advantages of the technology with the efficiency of busbar design. These modern electrode patterns are expected to emerge into the mainstream electrodes that would be adopted for a wider range of applications, especially engineering and design.

**Fractals in Biometric Applications:** Fingerprints are the simplest and most reliable biometric features that are widely used for identification purposes. Fingerprints exhibit self-similarity at multiple scales, and a fingerprint database can be classified using fractal dimension, but a fingerprint cannot be identified with fractal dimension uniquely. In [66], a novel Fingerprint Fractal Identification System (FFIS) was presented for identifying a fingerprint uniquely using fractal geometry and game theoretic techniques.

## 8. Conclusions

This article presents a comprehensive survey of fractals with focus on their applications in innovative and emerging research fields. With this extensive survey, we have tried to demonstrate the importance of fractals in engineering, industry and commercial applications by considering fractals in the design of fractal antennas, image processing, landscape generation, and fracture mechanics. Some future-ready applications of fractals are also discussed toward the end. In Part I of this survey of fractals [1], we considered the mathematics of fractals using iterated function systems, attractors, fractal dimensions, etc. and their appearance in fractal arts, ceramic products, fractal clothing and in fractal tilings.

Fractals have been studied in mathematics, computer science, engineering, physics, chemistry, biology, geology, social science, economics, technology, art, architecture and many other areas. Fractals have deep relevance in chaos theory because the graphs of most chaotic processes are fractals. The field of fractals has enormous potential to expand and take hold into many evolving areas of research, and even a voluminous book would be inadequate to discuss all of these in one place.

In summary, fractal geometry is the language of nature, and Benoît Mandelbrot has given us a new science which is applicable almost everywhere with an mind-opening effect on everyone who has come across it. This new language is changing our scientific world rapidly with sustainable solutions.

We close with a remark by Mandelbrot from the book *The Fractalist. Memoirs of a Scientific Maverick*, which is an inspirational collection of his own reflections and thoughts.

*"Within the sciences of nature, I was a pioneer in the study of familiar shapes, like mountains, coastlines, clouds, turbulent eddies, galaxy clusters, trees, the weather, and others beyond counting".*

*Benoît B. Mandelbrot, (2010)*

**Author Contributions:** Conceptualization, A.H. and M.S.; methodology, A.H., M.S. and M.N.N.; software, M.N.N. and M.S.C.; validation, A.H., M.S. and M.N.N.; formal analysis, A.H.; investigation, M.N.N., M.S.C. and M.S.; resources, M.N.N. and M.S.C.; writing—original draft preparation, A.H., M.N.N. and M.S.C.; writing—review and editing, A.H., M.N.N., M.S.C. and M.S.; supervision, A.H. and M.S. All authors have read and agreed to the published version of the manuscript.

**Funding:** This research received no external funding.

**Institutional Review Board Statement:** Not applicable.

**Informed Consent Statement:** Not applicable.

**Data Availability Statement:** Not applicable.

**Acknowledgments:** The authors are thankful to the referees for their careful reading of the manuscript and for giving valuable suggestions to improve it.

**Conflicts of Interest:** The authors declare no conflict of interest.

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
