# Peer review of "Fractals: An Eclectic Survey, Part II"

_fractalfract, doi:10.3390/fractalfract6070379_

Round 1

Reviewer 1 Report

Authors make an eclectic survey of fractals describing the infinite complexity and amazing beauty of fractals from historical, theoretical, mathematical, aesthetical, and technological aspects including their diverse applications in various fields. In this article, their focus is on engineering, industrial, commercial and futuristic applications of fractals. Such a type of research corresponds to the aims and scope of the journal. 

There are the following comments:

1.     The abstract should be expanded approximately 2-2.5 times.

2.     More references should be added in the introduction to show the previous work in this area.

3.     In the introduction, it is necessary to present more clearly the novelty of the results obtained including the author’s contribution to this problem area.

4.     Formulas everywhere must end with a comma or a dot.

5.     Reference to the theorems should be added.

6.      I also strongly recommend submitting a revised version using the MDPI template.

7.     There are some spelling mistakes. Please review the paper carefully.

Overall, I see that the paper is suitable for publishing after making the above comments

Reviewer 2 Report

This paper is a comprehensive survey of fractals with a focus on their applications in engineering, industry, and commercial applications. The proposed parts are given in detail and appropriate references are provided by the authors. 

Reviewer 3 Report

The authors give a wonderful introductionary survey over a broad range of different emerging applications of fractal geometries (mostly 1D to 2D structures) in engineering, industry and futuristic fields.

They perfectly fulfill their claim of allowing a broad class of reader to follow the manuscript but they also give a quite impressive list of in depth papers for subsequent investigations.

There could be some minor change done for a better readability:

1) Table 1: Please use indents in the pseudo code for the interlaced loops.

2) Section 3.2-3.3: I somehow got confused due to the use of the dimensions in this two sections. 

Figure 7(b) shows a 1D-landscape. Figure 8 shows 1D-landscapes for different H, but in the caption 2D-landscape is written. Figure 9 gives pictures of 2D-landscapes although in sec. 3.3 it is written that 2D-landscape is in the xy-plane with displacement in z what means, that they are 3D. This would fit better to figure 11 but they are an extension of the 2D-mid-point-displacement method to 3D.

Thus, in these section a clarification would be helpful.

Figure 8: 

- caption: Please check, whether it is a 1D or 2D landscape?

- Please add axis labels to the figures.

Figure 9:

- What is the white circle good for?

- Due to the caption it sounds like the right figure only shows one iteration step with exponential decay. For a better comparison of both figures the same number of iteration steps might be better.

3) sec. 3.4: Please give for the parameter H the definition in order to avoid misunderstands with the definition of H for the midpoint-displacement method.

And there are some minor typos:

p.11, sec. 4.2.3: fl=Lower -> f_l=Lower

p.13, pre-last parag.: reference to fig. 15(c) didn't work.

p.29: index of K_{eff} should be straight font.

Reviewer 4 Report

Publishable after revisions of the English and style of writing
